# eNOS-induced vascular barrier disruption in retinopathy by c-Src activation and tyrosine phosphorylation of VE-cadherin

Takeshi Ninchoji[1†]*, Dominic T Love[1†], Ross O Smith[1], Marie Hedlund[1], Dietmar Vestweber[2], William C Sessa[3], Lena Claesson-Welsh[1]*

[1]Uppsala University, Rudbeck Laboratory, Department of Immunology, Genetics and Pathology, Uppsala, Sweden; [2]Max Planck Institute for Molecular Biomedicine, Münster, Germany; [3]Yale University School of Medicine, Department of Pharmacology and Vascular Biology and Therapeutics Program, New Haven, United States

## Abstract

**Background:** Hypoxia and consequent production of vascular endothelial growth factor A (VEGFA) promote blood vessel leakiness and edema in ocular diseases. Anti-VEGFA therapeutics may aggravate hypoxia; therefore, therapy development is needed.

**Methods:** Oxygen-induced retinopathy was used as a model to test the role of nitric oxide (NO) in pathological neovascularization and vessel permeability. Suppression of NO formation was achieved chemically using L-NMMA, or genetically, in endothelial NO synthase serine to alanine (S1176A) mutant mice.

**Results:** Suppression of NO formation resulted in reduced retinal neoangiogenesis. Remaining vascular tufts exhibited reduced vascular leakage through stabilized endothelial adherens junctions, manifested as reduced phosphorylation of vascular endothelial (VE)-cadherin Y685 in a c-Src-dependent manner. Treatment with a single dose of L-NMMA in established retinopathy restored the vascular barrier and prevented leakage.

**Conclusions:** We conclude that NO destabilizes adheren junctions, resulting in vascular hyperpermeability, by converging with the VEGFA/VEGFR2/c-Src/VE-cadherin pathway.

**Funding:** This study was supported by the Swedish Cancer foundation (19 0119 Pj ), the Swedish Research Council (2020-01349), the Knut and Alice Wallenberg foundation (KAW 2020.0057) and a Fondation Leducq Transatlantic Network of Excellence Grant in Neurovascular Disease (17 CVD 03). KAW also supported LCW with a Wallenberg Scholar grant (2015.0275). WCS was supported by Grants R35 HL139945, P01 HL1070205, AHA MERIT Award. DV was supported by grants from the Deutsche Forschungsgemeinschaft, SFB1450, B03, and CRU342, P2.

*For correspondence: nincho830@gmail.com (TN); lena.welsh@igp.uu.se (LC-W)

†These authors contributed equally to this work

**Competing interests:** The authors declare that no competing interests exist.

## Introduction

Pathological neovascularization is intimately associated with the progression of several retinal diseases, including retinopathy of prematurity, diabetic retinopathy, and exudative age-related macular degeneration. Neovascularization occurs in response to hypoxia, tissue ischemia, and the consequent production of angiogenic agonists, such as vascular endothelial growth factor A (VEGFA), a potent inducer of vessel formation and vascular leakage (*Campochiaro, 2015*; *Semenza, 2012*). The new vessels formed during retinal ischemia are often dysfunctional and fail to stabilize (*Fruttiger, 2007*; *Krock et al., 2011*), leading to vessel leakage or hemorrhaging and to retinal detachment, visual impairment, and even blindness. Therefore, suppression of neoangiogenesis, and

thereby, retinal edema, is a therapeutic goal in the treatment of ischemic eye diseases (*Daruich et al., 2018*).

A number of therapeutic options designed to neutralize VEGFA by preventing binding to its receptor, VEGF receptor 2 (VEGFR2), such as bevacizumab, ranibizumab, and aflibercept, decrease neovascular formation as well as edema (*Mintz-Hittner et al., 2011*). However, they do not correct the underlying hypoxia; in fact, vessel regression may instead further aggravate hypoxia. Moreover, in many cases, anti-VEGF therapies can induce an elevation in intraocular pressure and hemorrhaging (*Diabetic Retinopathy Clinical Research Network et al., 2015*). The repeated intravitreal injections of anti-VEGF-therapy present a potential for infections and scarring (*Patel et al., 2013*); in addition, side effects including disrupted neural development in infants have been reported (*Canadian Neonatal Network and the Canadian Neonatal Follow-Up Network Investigators et al., 2016*). Thus, even though the current therapy improves vision for many patients in the early phases of disease, there is a clear need for developing new treatments to suppress proangiogenic stimuli while also achieving a long-lasting effect with safe administration.

Angiogenesis and vascular permeability in the retina are initiated by the VEGFA/VEGFR2 signaling pathway. VEGFR2 is found not only on blood vascular endothelial cells but also on neuronal cells in the retina, likely contributing to the side effect of VEGFA/VEGFR2 suppression on neural development in premature infants (*Nishijima et al., 2007*). VEGFR2 activity is initiated through VEGFA-induced receptor dimerization, kinase activation, phosphorylation of tyrosine residues in the receptor intracellular domain, and activation of signaling pathways. The VEGFR2 phosphotyrosine-initiated signaling pathways are now being unraveled. Phosphorylation of Y1212 in VEGFR2 is required for activation of phosphatidyl inositol three kinase and AKT (*Testini et al., 2019*), while phosphorylation of Y949 is required for activation of the c-Src pathway and regulation of vascular permeability through vascular endothelial (VE)-cadherin (*Li et al., 2016*). VE-cadherin is the main component of endothelial adherens junctions, strongly implicated in regulation of vascular permeability, leakage, and associated edema (*Giannotta et al., 2013*). In particular, phosphorylation of the Y685 residue in VE-cadherin correlates with VEGFA-induced vascular hyperpermeability (*Smith et al., 2020*; *Wessel et al., 2014*).

Endothelial nitric oxide synthase (eNOS), activated downstream of VEGFA through phosphorylation on S1177 (S1176 in mice) by AKT (*Fulton et al., 1999*), produces nitric oxide (NO) and regulates vascular permeability. Both mice with a constitutive eNOS gene (*Nos3*) inactivation (*Fukumura et al., 2001*) and mice expressing an eNOS serine to alanine point mutation (S1176A) (*Nos3$^{S1176A/S1176A}$*) (*Di Lorenzo et al., 2013*) show reduced extravasation of bovine serum albumin in the healthy skin in response to VEGFA challenge. An important mediator of the effect of eNOS-generated NO is the relaxation of peri-vascular smooth muscle cells (vSMCs), leading to increased vessel diameter and enhanced blood flow and thereby flow-driven vascular sieving (*Sessa, 2004*). In addition, eNOS activity correlates with tyrosine phosphorylation of VE-cadherin in cultured endothelial cells (*Di Lorenzo et al., 2013*), providing a mechanism for how eNOS activity may directly affect vascular permeability, distinct from vasodilation.

In contrast to the skin vasculature, the healthy retinal vasculature is protected by a stringent blood-retinal barrier. In retinal diseases, the barrier is disrupted, leading to increased vascular permeability (*Zhao et al., 2015*). In accordance, NO and related reactive oxygen species (ROS) are important pathogenic agents in retinopathy (*Opatrilova et al., 2018*). However, the molecular mechanisms whereby eNOS/NO interferes with the retinal vascular barrier and contributes to pathological vascular permeability in eye disease have remained unexplored.

Here, we show that suppressed NO formation via the use of the competitive NOS inhibitor, Nω-Methyl-L-arginine (L-NMMA), which inhibits NO formation from all three NOS variants (eNOS, inducible NOS, and neuronal NOS), or an eNOS mutant, S1176A, negates neovascular tuft formation and vascular leakage during retinal disease. Mechanistically, NO promotes c-Src Y418 phosphorylation at endothelial junctions and phosphorylation of VE-cadherin at Y685, required for dismantling of adherens junctions. Inhibition of NO formation by L-NMMA treatment suppresses vascular leakage also from established neovascular tufts, separating regulation of leakage from the angiogenic process as such. Mice expressing a VE-cadherin tyrosine to phenylalanine mutation (VEC-Y685F) are resistant to eNOS inhibition, supporting the model that NO regulates adherens junctions through direct effects on c-Src and VE-cadherin. These data suggest that eNOS/NO promote vascular permeability not only through the established effect on vSMC relaxation and increased flow-driven permeability to

solute and small molecules in the precapillary arterial bed, but also through disruption of adherens junctions allowing leakage of larger molecules from the postcapillary venular bed.

# Materials and methods

## Key resources table

| Reagent type (species) or resource | Designation | Source or reference | Identifiers | Additional information |
|---|---|---|---|---|
| Strain; strain background (*Mus musculus*) | $Nos3^{+/+}$ (C57BL/6J) | DOI:10.1016/j.bbrc.2012.12.110 | | |
| Strain; strain background (*Mus musculus*) | $Nos3^{S1176A/S1176A}$ (C57BL/6J) | DOI:10.1016/j.bbrc.2012.12.110 | | |
| Strain; strain background (*Mus musculus*) | *Cdh5*-WT (C57BL/6J) | DOI: 10.1038/ni.2824 | | Referred to as VEC-WT throughout |
| Strain; strain background (*Mus musculus*) | *Cdh5*-Y685F (C57BL/6J) | DOI: 10.1038/ni.2824 | | Referred to as VEC-Y685F throughout |
| Cell line (*Homo-sapiens*) | Human retinal microvascular endothelial | Cell Systems | Cat# ACBRI 181 | Primary cells HRMEC |
| Antibody | Anti-VE-cadherin pY685 (rabbit polyclonal) | DOI:10.1038/ncomms2199 | | IF (1:50), WB (1:1000) |
| Antibody | Anti-VE-cadherin (goat polyclonal) | R and D systems | Cat# AF1002 RRID:AB_2077789 | IF (1:200), WB (1:1000) |
| Antibody | Anti-eNOS (mouse monoclonal) | Abcam | Cat# ab76198 RRID:AB_1310183 | WB (1:1000) |
| Antibody | Anti-Src GD11 clone (mouse monoclonal) | Merck Millipore | Cat# 05–184 RRID:AB_2302631 | IF (1:200), WB (1:1000) |
| Antibody | Anti-c-Src pY418 (rabbit polyclonal) | Invitrogen | Cat# 44–660G RRID:AB_1500523 | IF (1:100), WB (1:1000) |
| Antibody | Anti-α-tubulin (mouse monoclonal) | Sigma-Aldrich | Cat# T9026 RRID:AB_477593 | WB (1:1000) |
| Antibody | Anti-eNOS pS1177 (mouse monoclonal) | BD Biosciences | Cat# 612392 RRID:AB_399750 | WB (1:1000) |
| Antibody | Anti-Akt (rabbit polyclonal) | Cell Signaling | Cat# 9272S RRID:AB_329827 | WB (1:1000) |
| Antibody | Anti-Akt pS473 (rabbit monoclonal) | Cell Signaling | Cat# 4058S RRID:AB_331168 | WB (1:1000) |
| Antibody | Anti-CD31 (rat monoclonal) | BD Biosciences | Cat# 553370 RRID:AB_394816 | IF (1:200) |
| Antibody | Anti-CD68 FA-11 clone (rat monoclonal) | BIO-RAD | Cat# MCA1957 RRID:AB_322219 | IF (1:300) |
| Antibody | Anti-NG2 (rabbit polyclonal) | Merck Millipore | Cat# AB5320 RRID:AB_91789 | IF (1:300) |
| Antibody | Anti-ERG (rabbit monoclonal) | Abcam | Cat# Ab92513 RRID:AB_2630401 | IF (1:200) |
| Antibody | Donkey anti-Rabbit IgG | ThermoFisher Scientific | Cat# A-31572 RRID:AB_162543 | IF (1:500) |
| Antibody | Donkey anti-Rat IgG | ThermoFisher Scientific | Cat# A-21208 RRID:AB_141709 | IF (1:500) |
| Antibody | Donkey anti-Goat IgG | ThermoFisher Scientific | Cat# A-11055 RRID:AB_2534102 | IF (1:500) |
| Antibody | Donkey anti-Mouse IgG, (H + L) HRP | ThermoFisher Scientific | Cat# A-16011 RRID:AB_2534685 | WB (1:10000) |

*Continued on next page*

*Continued*

| Reagent type (species) or resource | Designation | Source or reference | Identifiers | Additional information |
|---|---|---|---|---|
| Antibody | Donkey anti-Rabbit IgG, (H + L) HRP | ThermoFisher Scientific | Cat# A-16023 RRID:AB_2534697 | WB (1:10000) |
| Antibody | Donkey anti-Goat IgG, (H + L) HRP | ThermoFisher Scientific | Cat# A-15999 RRID:AB_2534673 | WB (1:10000) |
| Commercial assay, kit | Griess assay (nitrate/nitrite colorimetric assay kit) | Cayman Chemical | Cat# 780001 | |
| Commercial assay, kit | CD31 microbeads, mouse | Miltenyi Biotec | Cat# 130-097-418 | |
| Commercial assay, kit | RNeasy Mini Kit | QIAGEN | Cat# 74104 | |
| Chemical compound, drug | Nω-Methyl-L-arginine acetate salt (L-NMMA) | Sigma-Aldrich | Cat# M7033 | |
| Commercial assay, kit | Amersham ECL Prime Western Blotting Detection | GE Healthcare | Cat# RPN2232 | |
| Sequence-based reagent | *Nos3* forward | ThermoFisher Scientific | PCR primers | AAGGTGATGAGGACTCTGTGGC |
| Sequence-based reagent | *Nos3* reverse | ThermoFisher Scientific | PCR primers | GATATCTCGGGCAGCAGCTT |
| Sequence-based reagent | *Nos2* forward | ThermoFisher Scientific | PCR primers | GGTGAAGGGACTGAGCTGTTA |
| Sequence-based reagent | *Nos2* reverse | ThermoFisher Scientific | PCR primers | TGAGAACAGCACAAGGGGTTT |
| Sequence-based reagent | *Vegfa* forward | ThermoFisher Scientific | PCR primers | GCACATAGAGAGAATGAGCTTCC |
| Sequence-based reagent | *Vegfa* reverse | ThermoFisher Scientific | PCR primers | CTCCGCTCTGAACAAGGCT |
| Sequence-based reagent | *Tbp* forward | ThermoFisher Scientific | PCR primers | CCTTGTACCCTTCACCAATGAC |
| Sequence-based reagent | *Tbp* reverse | ThermoFisher Scientific | PCR primers | ACAGCCAAGATTCACGGTAGA |
| Sequence-based reagent | *Ubc* forward | ThermoFisher Scientific | PCR primers | CCCACACAAAGCCCCTCAAT |
| Sequence-based reagent | *Ubc* reverse | ThermoFisher Scientific | PCR primers | AAGATCTGCATCGTCTCTCTCAC |
| Peptide, recombinant protein | VEGFA, recombinant, mouse | Peprotech | Cat# 450–32 | |
| Commercial assay, kit | Duolink In Situ PLA Probe anti-Rabbit MINUS | Sigma-Aldrich | Cat# DUO92005 RID:AB_2810942 | |
| Commercial assay, kit | Duolink In Situ PLA Probe anti-Mouse PLUS | Sigma-Aldrich | Cat# DUO92001 RRID:AB_2810939 | |
| Commercial assay, kit | Duolink In Situ Detection Reagent (Orange) | Sigma-Aldrich | Cat# DUO92007 | |
| Other | SuperScript III Reverse Transcriptase | Invitrogen | Cat# 18080093 | |
| Other | DAF-FM diacetate (DA) | Sigma-Aldrich | Cat# D1946-1MG | |

*Continued on next page*

*Continued*

| Reagent type (species) or resource | Designation | Source or reference | Identifiers | Additional information |
|---|---|---|---|---|
| Other | Lycopersicon Esculentum (Tomato) Lectin (LEL, TL), Fluorescein | Vector Laboratories | Cat# FL-1171–1 | |
| Other | Fluoro-Max Dyed Green Aqueous Fluorescent Particles | ThermoFisher Scientific | Cat# G25 | |
| Other | Hoechst 33342 | ThermoFisher Scientific | Cat# H3570 | IF (1:1000) |
| Other | Alexa Fluor 488-Isolectin B4 | Sigma-Aldrich | Cat# I21411 RRID:AB_2314662 | IF (1:500) |
| Other | Alexa Fluor 647-Isolectin B4 | Sigma-Aldrich | Cat# I32450 RRID:SCR_014365 | IF (1:500) |
| Software | ImageJ | NIH, Bethesda, MD | RRID:SCR_003070 | |
| Software | GraphPad Prism | GraphPad | RRID:SCR_002798 | |

## Animal studies

*Nos3$^{S1176A/S1176A}$* mice on a C57Bl/6J background have been described (**Kashiwagi et al., 2013**). *Cdh5*-WT and *Cdh5-Y685F* mice (referred to as VEC-WT and VEC-Y685F throughout text), also on C57Bl/6J background, were generated using a wild-type murine VE-cadherin/*Cdh5* cDNA construct (VEC-WT) or a mutant cDNA in which Y685 is replaced by F685 (VEC-Y685F, **Wessel et al., 2014**). Both strains were maintained by crossing heterozygous mice. Wild-type C57BL/6J mice (Jackson Laboratory) and the Y685F strain were treated, when indicated, with Nω-methyl-L-arginine acetate salt (L-NMMA; Sigma-Aldrich) in PBS, 20 µg/g body weight, by intraperitoneal injection from postnatal (P) day 12 to P16.

Wild-type mice were also treated, when indicated, with a single dose of L-NMMA in PBS, 60 µg/g body weight, by intraperitoneal injection on P16.

Mouse husbandry and OIR-challenge took place at Uppsala University, and the Local Ethics committee approved all animal work for these studies. Animal handling was in accordance with the ARVO statement for the Use of Animals in Ophthalmologic and Vision Research. All animal experiments were repeated individually at least three times (biological repeats).

## Oxygen-induced retinopathy

A standard oxygen-induced retinopathy (OIR) model was used (**Connor et al., 2009**). Briefly, each litter of pups was placed, along with the mother, into a chamber maintaining 75.0% oxygen (ProOx 110 sensor and AChamber, Biospherix, Parish, NY) from P7 to P12. They were then returned to normal atmosphere (~21% oxygen), until P17 (termination). The lactating mother was removed each day, P8–P11, and placed in normal atmosphere for 2 hr, to prevent oxygen toxicity. At P17, pups were weighed and sacrificed. Eyes were enucleated and fixed in 4% paraformaldehyde (PFA) at room temperature for 30 min. See *Supplementary file 1A* for data on body weights at P17 after OIR. No mice were excluded from analysis.

## Quantification of avascular area and neovascular tufts

Avascular area and neovascular tuft formation were determined by immunostaining retinas followed by imaging (Leica SP8 confocal microscope) and analysis. Quantification of total vascularized area, central avascular area, and tuft area was performed by outlining images manually in ImageJ (NIH, Bethesda, MD). Using a tilescan of IB4 staining for the entire retina, the freehand selection tool was used to demarcate the vascular front, creating a region of interest (ROI) for the total vascularized area. The freehand selection tool was used to outline IB4-positive vessels from neovascular tufts (regions with disorganized dilated vessels). The ROIs for tufts were merged into a single ROI corresponding to the total neovascular tuft area for each retina. The tuft area normalized to the total

vascularized area was reported as a percentage of the total retina that contained tufts. Similarly, the avascular region was determined using the freehand selection tool to outline the central avascular regions. Regions where the superficial layer of capillaries was absent were determined and merged to form a single ROI corresponding to the entire avascular region for each retina. The avascular area normalized to the total vascularized area was reported as a percentage of the total retina that was still avascular. The researcher was blinded to the genotype of the sample when performing quantifications.

## Microsphere assay

For microsphere extravasation experiments, mice at P17 were briefly warmed under heat lamp to dilate tail veins before injection of microspheres (1% solution of 25 nm fluorescent microspheres; 50 μl per mouse into the tail vein; ThermoFisher). Microspheres circulated for 15 min. To remove blood and microspheres from the retinal vessels, mice were perfused with room temperature phosphate-buffered saline (PBS) containing $Ca^{2+}/Mg^{2+}$, using a peristaltic pump for 2 min, under full isoflurane anesthesia. The eyes were then enucleated and fixed in 4% PFA at room temperature for 30 min before dissection, IB4 staining, and mounting for microscopy (Leica SP8 confocal microscope, 63× objective).

Using ImageJ software, the microsphere channel and IB4-vessel channel (488 for Green fluorescence and 647 for IB4) were adjusted with threshold (Huang for IB4 and Triangle for FITC) for each channel. Extravasated microsphere area was calculated by measuring the signal in the green fluorescence channel after removing any signal contained within the ROI corresponding to the IB4-positive area. The Analyze Particles function was employed to quantify the microspheres. A lower limit of 10 pixels was selected to distinguish the microsphere signal from background noise. The mean area density for each group of mice was calculated from the median value of all images of the eyes of each mouse (*Smith et al., 2020*). To quantify leakage based on microscopic images, the amount of tracer extravasation was normalized to blood vessel density. The researcher was blinded to the genotype of the sample when performing quantifications. See *Supplementary file 1B* for data on body weights at P17; microsphere analyses. No mice were excluded from analysis.

## Quantification of perfusion, pericyte coverage, and macrophage infiltration

Mice at P17 were anesthetized with an intraperitoneal injection of ketamine/xylazine (100 μg/g and 10 μg/g bodyweight). Once sedated, the mice were given a retro-orbital injection of 25 μl tomato lectin (25 mg/mL) and allowed to circulate for 5 min, followed by perfusion with room temperature PBS containing $Ca^{2+}/Mg^{2+}$, using a peristaltic pump for 2 min. The eyes were then enucleated and fixed in 1% paraformaldehyde (PFA) at room temperature for 30 min before dissection. The eyes were processed for detection of IB4, inflammatory cells (CD68+), and pericytes (NG2+) and mounted for microscopy (Leica SP8 confocal microscope, 20× objective).

Using imageJ software, macrophage infiltration was analyzed by counting the total number of CD68-positive particles in the entire retina. Pericyte coverage was analyzed by normalizing the NG2-positive area to the IB4 area of individual tufts. Perfusion was determined by normalizing the tomato lectin positive area to the IB4 area of individual tufts. The researcher was blinded to the genotype of the sample when performing quantifications. See *Supplementary file 1C* for data on body weights at P17 after OIR for inflammation, pericyte analysis, and perfusion. No mice were excluded from analysis.

## Endothelial cell isolation from mouse lung

Mouse lungs were harvested from pups (age; P8–P10), minced, and digested in 2 mg/mL collagenase type I (385 U/mg; Worthington) in PBS with $Ca^{2+}/Mg^{2+}$ for 1 hr at 37°C. Cells were then isolated using CD31 Microbeads and MACS cell isolation equipment and reagents (Miltenyi Biotec). The cells were seeded at $3 \times 10^5$ cells/mL and cultured in MV2 medium with supplements and serum (PromoCell).

## Griess reagent assay

Isolated endothelial cells were seeded at $3 \times 10^5$ cells/mL into 6 cm cell culture plates and allowed to adhere at 37°C and 5% $CO_2$ overnight. After 24 hr a Griess Assay was performed (nitrate/nitrite colorimetric assay, Cayman chemical) according to the manufacturer's instruction. Once complete, the cells were lysed in 1% (wt/vol) NP 40, 25 mM Tris–HCl pH 7.6, 0.1% SDS, 0.15 M NaCl, 1% sodium deoxycholate, and 1× Protease Inhibitor Cocktail (Roche), and concentration of nitrate/nitrite was normalized against protein concentration, measured using the BCA protein detection kit (ThermoFisher).

## Proximity ligation assay

Isolated endothelial cells, serum starved at 37°C in MV2 medium (containing no growth factors) 3 hr before stimulation with murine $VEGFA_{164}$ (50 ng/mL; R and D Systems), followed by fixation in 3% PFA for 3 min, permeabilized in 0.1% Triton X-100 for 3 min, and postfixed in 3% PFA for 15 min. Samples were blocked in Duolink blocking buffer for 2 hr at 37°C and used for proximity ligation assay (PLA). The Duolink protocol (Sigma-Aldrich) was followed using anti-phospho-Src Tyr 418 (Invitrogen) and total c-Src (Merck Millipore) antibodies, and oligonucleotide-linked secondary antibodies, denoted PLUS and MINUS probes, followed by the detection of reactions with fluorescent probes. Upon completion of the PLA protocol, cells were counterstained with antibodies against VE-cadherin (R and D systems), and Hoechst 33342 (ThermoFisher) to detect nuclei. Only cells positive for VE-cadherin were imaged and analyzed. To determine c-Src p418 association with VE-cadherin, a mask of the VE-cadherin channel was created and only points that aligned completely within the VE-cadherin mask were counted and expressed against the area of VE-cadherin per field of view (ImageJ software, NIH). As a technical control for each experiment, the same procedure was performed with the omission of either of the primary antibodies or the PLUS/MINUS probes.

## Immunofluorescent staining

Whole mount immunostaining was performed on PFA-fixed retinas incubated in blocking buffer for 2 hr (Buffer b; bovine serum albumin [BSA]/2% fetal calf serum [FCS]/0.05% Na-deoxycholate/0.5% Triton X-100/0.02% Na Azide in PBS). Incubation with primary antibodies over night at 4°C on a rocking platform was followed by incubation with secondary antibodies overnight at 4°C. Retinas were mounted on slides using Fluormount G. Images were taken by Leica SP8 confocal microscope and acquired with the 10× or 63× objective. Processing and quantification of images was done with ImageJ software (NIH). Quantification in the retina of total vascularized area, central avascular area, and area covered by neovascular tufts was performed by outlining images manually in ImageJ. Avascular area and tuft area were normalized to the total vascularized area of the retina.

## Quantitative real-time PCR

RNA from retinas were purified using RNeasy Kit (Qiagen). One microgram of RNA was reverse transcribed using SuperScript III (Invitrogen), and quantitative PCR were assayed using *Mus musculus* primers against *Vegfa* (Mm00437306_m1, ThermoFisher), *Nos3* (Mm00435217_m1), and *Nos2* (Mm00440502_m1). The expression levels were normalized against TATA binding protein (TBP) *Mus musculus* (Mm01277042_m1, ThermoFisher) and Ubiqutin C (*UBC*) *Mus musculus* (Mm02525934_g1, ThermoFisher).

## Cell culture and treatment

Human retinal microvascular endothelial cells (HRMECs; Cell Systems, #ACBRI 181) were cultured in a complete classic medium kit with serum and CultureBoost (Cell Systems, #4Z0–500). The cells were supplied mycoplasma free. HRMECs were used and passaged in 10 cm cell culture plates coated with attachment factor, between passages 5–10 for all experiments. All cells were serum starved for 3 hr at 37°C in MV2 medium (containing no growth factors) before stimulation. Murine VEGFA (Peprotech) was used at 50 ng/mL for in vitro analyses. L-NMMA (1 mM in PBS) was administrated 1 hr before stimulation with VEGFA.

## Immunoblot

Cells were lysed in 1% (wt/vol) NP 40, 25 mM Tris–HCl pH 7.6, 0.1% SDS, 0.15 M NaCl, 1% sodium deoxycholate, $1\times$ Protease Inhibitor Cocktail (Roche), 1 mM $Na_3VO_4$ (Sigma), and centrifuged at 21,100 g for 10 min. Protein concentration was measured with the BCA protein detection kit (ThermoFisher). Proteins were separated on a 4–12% Bis–Tris polyacrylamide gel (Novex by Life Technologies) and transferred to an Immobilon-P PVDF membrane (Millipore) using the Criterion Blotter System (Bio–Rad). The membrane was blocked with 3–5% skim milk in Tris-buffered saline (TBS; 0.1% Tween). For phosphotyrosine antibodies, blocking was done in 5% BSA in TBS, 0.1% Tween. The membrane was incubated with first antibodies overnight at 4°C. Membranes were then washed in TBS, 0.1% Tween and incubated with horseradish peroxidase (HRP)-conjugated secondary anti-mouse antibody (1:10,000; Invitrogen) in 3–5% skim milk, respectively, followed by final wash in TBS, 0.1% Tween and development using ECL prime (GE Healthcare). Luminescence signal was detected by the ChemiDoc MP system (Bio–Rad) and densitometry performed using Image Lab software (ver 4, Bio-Rad).

## Antibodies

The retinal vasculature was stained with directly conjugated Alexa Fluor 488-Isolectin B4 (1:200; Sigma, I21411) or Alexa Fluor 647-Isolectin B4 (1:200; Sigma, I32450). EC junctions and phosphorylated VE-cadherin were detected with anti-VE-cadherin antibody (1:200, R and D, AF1002) and affinity purified rabbit antibodies against VE-cadherin pY685; a kind gift from Prof. Elisabetta Dejana, Uppsala University/IFOM Milano (Orsenigo et al., 2012). For PLAs, VE-cadherin was detected using goat anti-VE-cadherin antibody (1:200, R and D Systems, AF1002). c-Src was detected using rabbit anti-Src (GD11 clone) antibody (1:200, Merck Millipore, Mouse, 05–184). Phosphorylated c-Src was assessed using rabbit anti-Src pY418 antibody (1:100, Invitrogen, 44–660G). Nuclei were detected using Hoechst 33342 (1:1000, ThermoFisher, H3570). For immunoblotting, the following antibodies were used as primaries: mouse-anti-$\alpha$-tubulin (1:1000, Sigma, T9026), mouse anti-eNOS (1:1000, Abcam, ab76198), mouse anti-eNOS pS1177 (1:1000, BD, 612392), rabbit anti-Akt (1:1000, Cell Signaling, 9272S), rabbit anti-Akt pS473 (1:1000, Cell Signaling, 4058S), rabbit anti-Src (GD11 clone) antibody (1:1000, Merck Millipore, Mouse, 05–184), rabbit anti-Src pY418 antibody (1:1000, Invitrogen, 44–660G), goat anti-VE-cadherin (1:1000, R and D, AF1002), and rabbit anti-VE-cadherin pY685 (1:1000, DOI: 10.1038/ncomms2199). Detection: Amersham ECL prime Western blotting detection reagent (GE Healthcare, RPN2232).

## DAF-FM DA assay

Intracellular NO was measured in real time using the NO-specific fluorescence probe DAF-FM DA solution (Sigma-Aldrich). DAF-FM DA diffuses freely across the membrane and is hydrolyzed by intracellular esterases, resulting in the formation of DAF-FM. Intracellular DAF-FM reacts with the NO oxidation product $N_2O_2$, which generates the stable highly fluorescent derivative DAF-FM triazole. Cells were washed with modified HEPES buffer (20 mM HEPES buffer [Gibco] with 5 mM glucose, 50 µM L-arginine, and 0.1% BSA, pH 7.0–7.4), incubated with 5 µM DAF-FM DA in modified HEPES buffer for 30 min at room temperature, washed again, and finally incubated in modified HEPES buffer for 30 min at 37°C in the absence or presence of 1 mM L-NMMA. Fluorescence (emission wavelength, 485 nm; excitation wavelength, 538 nm) was measured at 37°C from 1 to 10 min using a fluorescence microtiter plate reader (Synergy HTX Multi-Mode Reader, BioTek). eNOS activity was expressed as the VEGFA-dependent increase in fluorescence per microgram of cellular protein. To determine the cellular protein content, the same cells were lysed in 1% (vol/vol) Triton X-100 and analyzed for protein content with the BCA protein detection kit. DAF-FM DA experiments were repeated three times. Within each experiment, four wells were used for each NO measurement.

## Statistical analysis

Statistical analysis was performed using GraphPad Prism 6 (GraphPad). An unpaired Student's t-test was used to compare means among two experimental groups. Two-way ANOVAs were performed when two factors were involved, for example, treatment and genotype. Multiple comparisons post hoc tests were chosen based on how many group comparisons were made. In all analyses, $p < 0.05$ was considered a statistically significant result. Values shown are the mean, with standard error of

the mean (S.E.M.) used as the dispersion measure. Biological replicates refer to individual mice/samples in a single experiment. Separate/individual experiments refer to experiments done at different times/days with independently generated material. A statistical method of sample size calculation was not used during the study design.

## Results

### Reduced neoangiogenic tuft formation in C57BL/6 OIR with suppressed NO formation

To determine whether eNOS inhibition affects pathological angiogenesis in retinopathy in C57BL/6J mice, pups were exposed to 75% oxygen from P7 to P12 (hyperoxic period) where after they were placed at normal, atmospheric conditions (21% oxygen; relative hypoxic period) from P12 to P17 (*Figure 1A*). During the P7–P12 hyperoxic period, VEGFA expression is suppressed, leading to endothelial cell death and avascularity in the superficial vessel layer (reviewed in *Scott and Fruttiger, 2010*). The relative decrease in oxygen concentration upon return to normal atmosphere at P12–P17 induces hypoxia-inducible factor-dependent gene regulation, causing oxygen-induced retinopathy (OIR) and the formation of neoangiogenic tufts (*Smith et al., 1994*) (see *Figure 1B* for schematic outline).

To specifically address the role of eNOS in vascular retinal disease, we used a genetic model in which eNOS S1176 (mouse numbering, *Fulton et al., 1999*; S1177 in human) is replaced by alanine (A) (*Kashiwagi et al., 2013*). Phosphorylation of eNOS on this serine residue is a prerequisite for eNOS-driven NO production, which was verified using a Griess assay on isolated endothelial cells from $Nos3^{+/+}$ and $Nos3^{S1176A/S1176A}$ mice (*Figure 1C*). Mice were subjected to the OIR regimen (*Figure 1D*). After OIR-challenge, $Nos3^{S1176A/S1176A}$ P17 retinas showed reduced pathological tuft area compared to $Nos3^{+/+}$ (*Figure 1E*), while the extent of avascularity was the same (*Figure 1F*). The average size of individual tufts was reduced in the $Nos3^{S1176A/S1176A}$ pups (*Figure 1G*), while the total number of tufts formed after OIR was unaffected (*Figure 1H*).

The suppressed formation of neoangiogenic tufts in the absence of eNOS S1176 phosphorylation was not due to developmental defects as $Nos3^{S1176A/S1176A}$ mice showed normal postnatal vascular development. Vascular plexus area and outgrowth in the retina, tip cell number and appearance, as well as branch points were all similar between the wild-type and the $Nos3^{S1176A/S1176A}$ retinas (*Figure 1—figure supplement 1A–G*). At P12 after OIR-challenge, there was also no difference in the degree of avascularity in the retina between $Nos3^{S1176A/S1176A}$ and $Nos3^{+/+}$ pups, indicating that the strains responded similarly to the hyperoxic challenge (*Figure 1—figure supplement 2A,B*). Moreover, no differences were found when comparing $Nos3^{+/+}$ and $Nos3^{S1176A/S1176A}$ retinas with regard to coverage of vessels by NG2-positive pericytes (*Figure 1—figure supplement 3A,B*) or the presence of CD68-positive macrophages (*Figure 1—figure supplement 3C,D*), at P17. However, vessel perfusion was enhanced in $Nos3^{S1176A/S1176A}$ tufts compared to the wild type (*Figure 1—figure supplement 3E,F*), most likely due to the smaller tufts formed in the $Nos3^{S1176A/S1176A}$ retinas.

Importantly, the reduced tuft area in the $Nos3^{S1176A/S1176A}$ condition was not a result of differences in $Nos3$ or $Nos2$ expression between the $Nos3^{+/+}$ and $Nos3^{S1176A/S1176A}$ mice before or after the OIR-challenge (*Figure 1—figure supplement 4A,B*). Reduced tuft area was also not a result of reduced VEGFA production as an equally induced level of VEGFA was seen in the mutant and wild-type retinas (*Figure 1—figure supplement 4C*). It should also be noted that the low relative expression level of $Nos2$ (encoding inducible nitric oxide synthase [iNOS]) compared to $Nos3$ (*Figure 1—figure supplement 4D,E*) emphasizes the primary role of eNOS as a source of endothelial NO, both in the unchallenged and OIR-treated condition.

### VEGFA induces eNOS phosphorylation and activity

VEGFA produced during the relative hypoxia phase (P12–P17) is an important instigator of edema in retinopathy (*Connor et al., 2009*; *Dor et al., 2001*). VEGFA/VEGFR2 signaling and vessel leakage correlates with phosphorylation of VE-cadherin on Y685 (*Li et al., 2016*; *Smith et al., 2020*; *Wessel et al., 2014*). The level of pY685 VE-cadherin was examined by immunostaining of $Nos3^{+/+}$ and $Nos3^{S1176A/S1176A}$ retinas at P17 after OIR-challenge. pY685 VE-cadherin immunostaining,

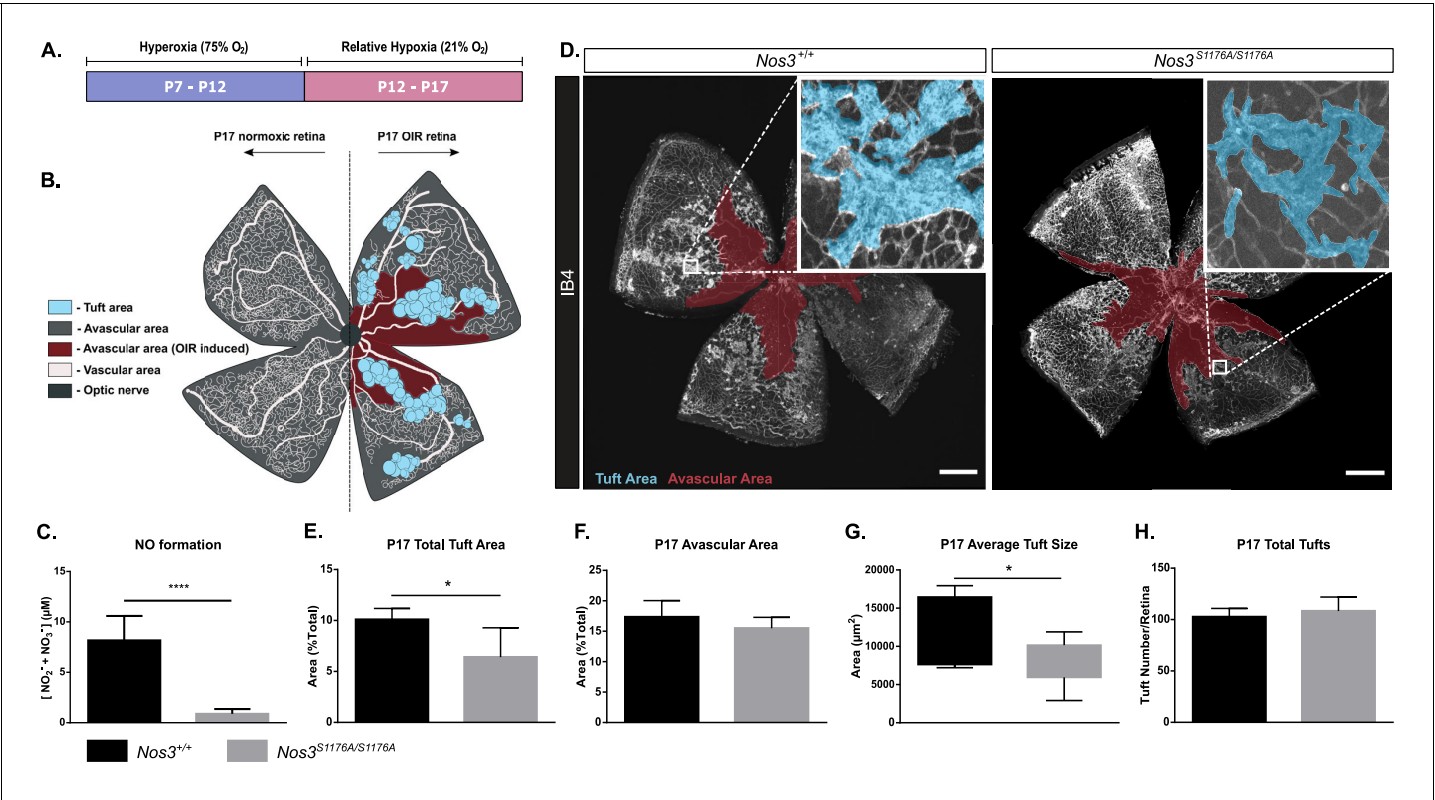

**Figure 1.** Suppressed tuft formation in the *Nos3^S1176A/S1176A* retina after OIR-challenge. (**A**) Outline of OIR-challenge protocol; pups were placed in 75% O₂ (hyperoxia) between P7 and P12, followed by return to normal atmosphere (relative hypoxia) until P17. (**B**) Schematic representation of vascular abnormalities after OIR in P17 retinas. (**C**) Nitric oxide formation determined using a Griess assay, expressed as the combined concentration of nitrite and nitrate, the end products of NO, reacting with molecules in biological fluids. Mean ± S.E.M. n = 3 mice/genotype. ****p<0.0001; t-test. (**D**) Representative images of whole mount retinas from *Nos3^+/+* and *Nos3^S1176A/S1176A* mice, collected at P17 after OIR-challenge, stained with isolectin B4 (IB4). Avascular area is marked in magenta and tufts in blue. Scale bar = 500 µm. (**E, F**) Tuft area (**E**) and avascular area (**F**) expressed as percentage of total vascular area at P17. (**G, H**) Tuft size in µm2 (**G**) and total number/FOV in P17 mice (**H**). For (**E–H**): mean ± S.E.M. n = 7 (*Nos3^+/+*) and 5 (*Nos3^S1176A/S1176A*) mice. *p<0.05; t-test.

The online version of this article includes the following source data and figure supplement(s) for figure 1:

**Source data 1.** Excel file containing numerical values collected from NO formation assays and OIR-induced tuft formation experiments in Nos3^+/+ and Nos3^S1176A/S1176A mice shown in *Figure 1*, *Figure 1—figure supplements 1–4*.

**Figure supplement 1.** Postnatal development of *Nos3^+/+* and *Nos3^S1176A/S1176A* retinal vasculature.

**Figure supplement 2.** Retina characteristics in *Nos3^+/+* and *Nos3^S1176A/S1176A* P12 pups.

**Figure supplement 3.** Pericytes, macrophage influx, and vascular perfusion in OIR-challenged *Nos3^+/+* and *Nos3^S1176A/S1176A* retinas.

**Figure supplement 4.** Expression of Nos2, Nos3, and Vegfa in *Nos3^+/+* and *Nos3^S1176A/S1176A* retinas.

normalized to the total VE-cadherin area, was significantly lower in *Nos3^S1176A/S1176A* tufts than in the WT tufts (*Figure 2A,B*).

We explored the potency of VEGFA/VEGFR2 in inducing NO generation and how this related to phosphorylation of VE-cadherin. eNOS was phosphorylated on S1177 in VEGFA-treated human retinal microvascular endothelial cells (HRMEC). Induction of eNOS phosphorylation appeared with similar kinetics but slightly more potently by VEGFA than by the inflammatory mediator histamine (*Figure 2—figure supplement 1A,B*; see *Figure 2—figure supplement 1C,D* for antibody validation), a well-known inducer of eNOS activity (*Thors et al., 2004*). eNOS phosphorylation was accompanied by NO production in response to VEGFA stimulation, as assessed using the fluorescent probe, DAF-FM DA added to the HRMEC culture medium. NO accumulated significantly by 1 min stimulation and still persisted at 10 min (*Figure 2—figure supplement 1E*). DAF-FM DA fluorescence, and therefore NO production, was blocked by incubating cells with L-NMMA (*Figure 2—figure supplement 1F*), to the level of the untreated control. Combined, these data show that VEGFA is a potent inducer of eNOS activity and NO production.

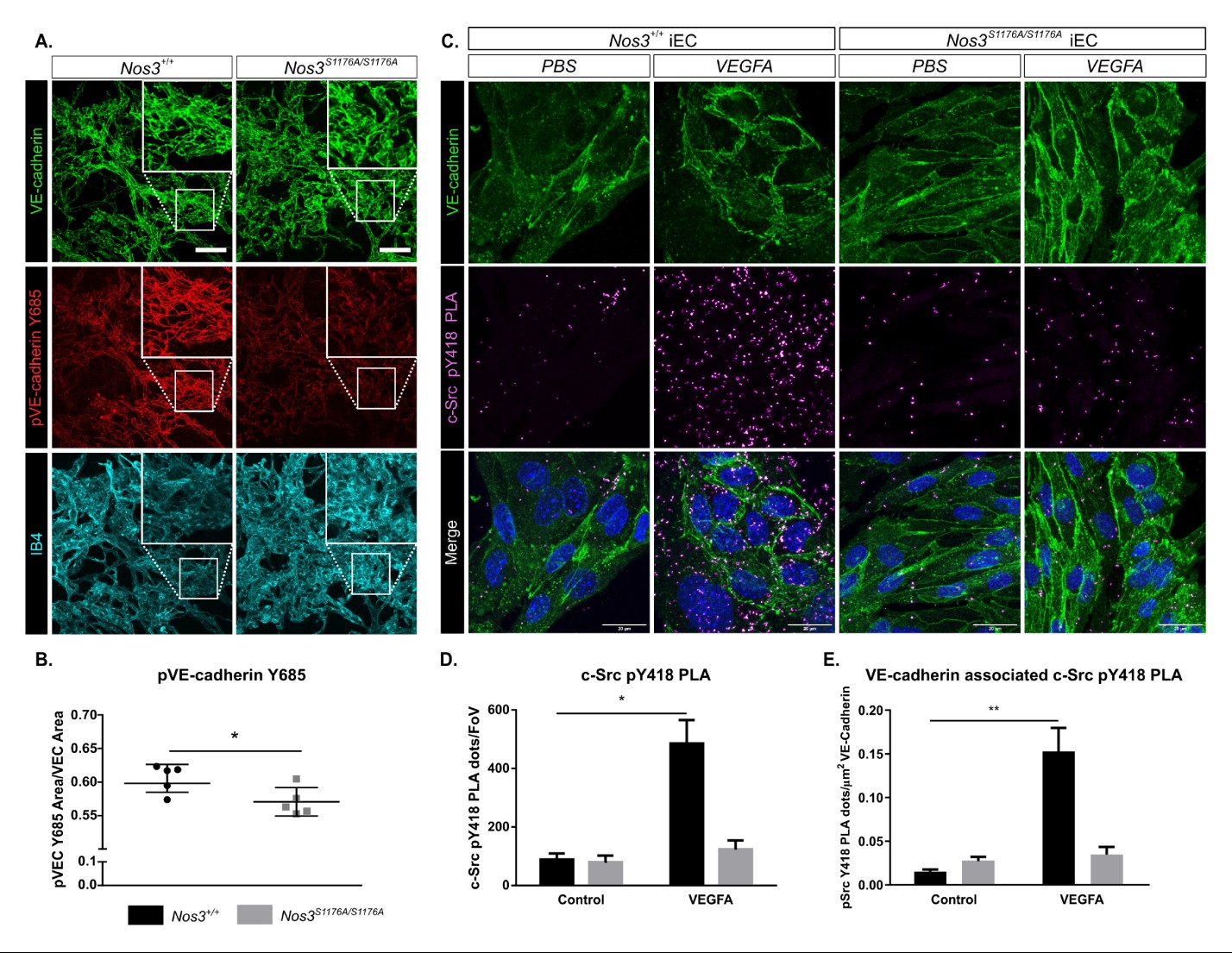

**Figure 2.** Suppressed c-Src Y418 and VE-cadherin Y685 phosphorylation in *Nos3^S1176A/S1176A* retinas and isolated endothelial cells. (**A**) Representative maximum intensity projections of similar sized tufts from *Nos3^+/+* and *Nos3^S1176A/S1176A* retinas showing VE-cadherin (green), pY685 VE-cadherin (red), and isolectin B4 (IB4; cyan). Scale bar = 50 μm. (**B**) Ratio of pY685 positive area/total VE-cadherin positive area. Mean ± S.E.M n = 3–6 images per group from 5 (*Nos3^+/+*) and 5 (*Nos3^S1176A/S1176A*) mice, three independent experiments, *p<0.05; t-test. (**C**) Representative images of VE-cadherin staining (green) and proximity ligation assay (PLA) to detect c-Src pY418 (magenta) in isolated mouse lung endothelial cells (iEC) from *Nos3^+/+* and *Nos3^S1176A/S1176A* mice. Scale bar = 20 μM. (**D**) c-Src pY418 PLA dots detected in PBS and VEGFA (50 ng/mL)-treated iECs from *Nos3^+/+* and *Nos3^S1176A/S1176A* mouse lungs. Data expressed as the number of dots per field of view. (**E**) c-Src pY418 PLA dots co-localized with VE-cadherin (green), normalized against total VE-cadherin area in the field of view. Mean ± S.E.M. Cells isolated from n = 4 (*Nos3^+/+*) and 4 (*Nos3^S1176A/S1176A*) mice, from three separate experiments. *, **p<0.05, 0.01; two-way ANOVA, Sidak's multiple comparisons test.

The online version of this article includes the following source data and figure supplement(s) for figure 2:

**Source data 1.** Excel file containing numerical values collected from biochemical analyses shown in *Figure 2*, *Figure 2—figure supplements 1–3*.
**Figure supplement 1.** VEGFA induced eNOS phosphorylation and activity in vitro.
**Figure supplement 2.** VEGFA induced phosphorylation of eNOS, AKT, VE-cadherin, and c-Src in vitro.
**Figure supplement 3.** c-Src pY418 immunostaining and PLA controls.

## VE-cadherin phosphorylation at Y685 is reduced in *Nos3^S1176A/S1176A* vessels after OIR due to the inhibition of c-Src Y418 phosphorylation

Phosphorylation of VE-cadherin on Y685 correlates with activation of the cytoplasmic tyrosine kinase c-Src (*Orsenigo et al., 2012*; *Wallez et al., 2007*). The NO-generating reagent, SNAP, can increase

the levels of activated c-Src phosphorylated on Y418 in fibroblast cultures (*Rahman et al., 2010*), indicating a potential role for NO in c-Src activation. We therefore tested whether eNOS 1176 phosphorylation correlates with phosphorylation of c-Src at Y418. In vitro, the cell-permeable NOS inhibitor L-NMMA had no effect on VEGFA-induced phosphorylation of eNOS (S1177). There was also no significant effect of L-NMMA treatment on VEGFA-induced phosphorylation of AKT (S473) (*Figure 2—figure supplement 2A–C*); VEGFA-mediated AKT activation is a prerequisite for phosphorylation of eNOS at S1177 (*Chen and Meyrick, 2004*; *Schleicher et al., 2009*). In contrast,

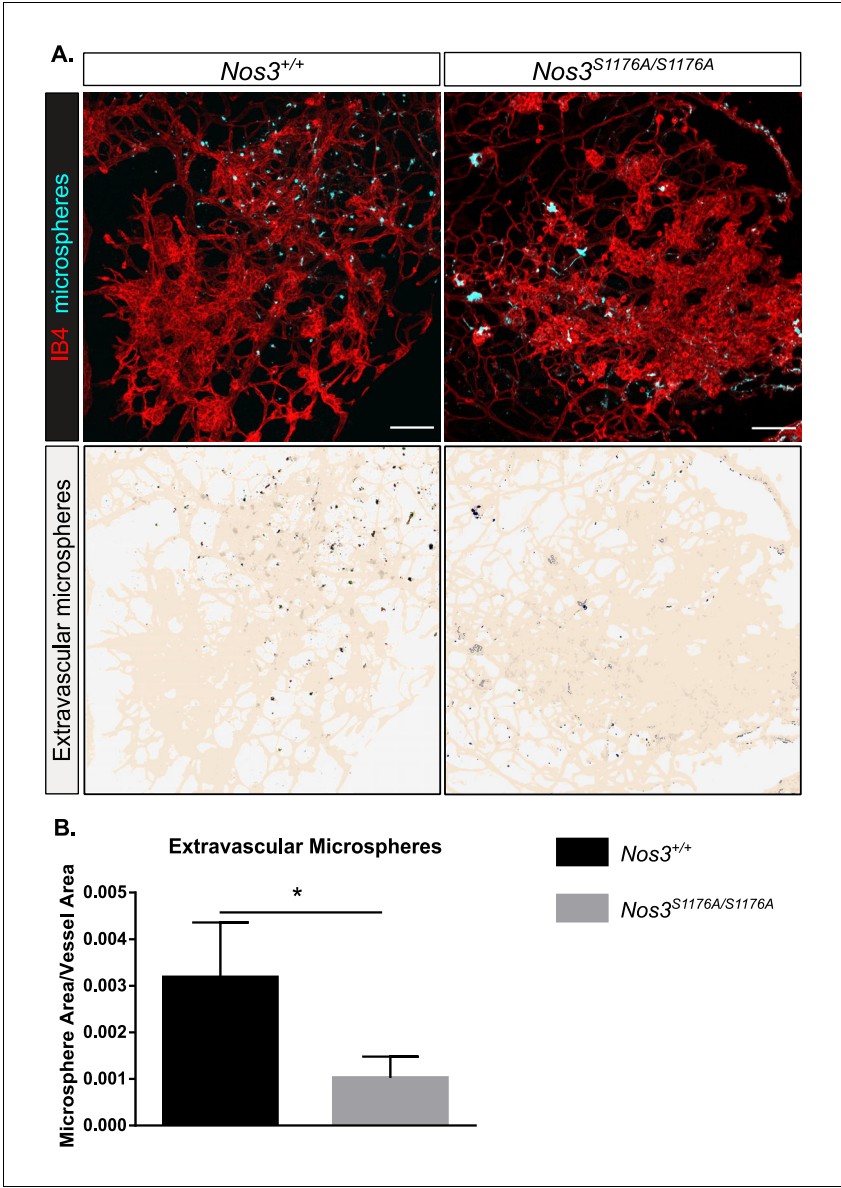

**Figure 3.** Suppressed microsphere leakage in *Nos3^{S1176A/S1176A}* retinas. (**A**) Representative images of similar sized tufts from *Nos3^{+/+}* and *Nos3^{S1176A/S1176A}* mice showing isolectin B4 (IB4; red), and leakage of tail-vein injected FITC-conjugated 25 nm microspheres (cyan) around the tufts. Scale bar = 100 μM. Lower panels show leakage maps. Dots that do not overlap with vessels (beige) are considered extravascular. (**B**) Quantification of the average area of extravascular microspheres normalized to IB4 area. Mean ± S.E.M. n = 4 (*Nos3^{+/+}*) and 4 (*Nos3^{S1176A/S1176A}*) mice, 6–15 images per mouse. *p<0.05, t-test.

The online version of this article includes the following source data for figure 3:

**Source data 1.** Excel file containing numerical values collected from microsphere leakage experiments from OIR retinas in Nos3^{+/+} and Nos3^{S1176A/S1176A} mice shown in *Figure 3*.

phosphorylation of VE-cadherin at Y685 was attenuated and delayed (*Figure 2—figure supplement 2D*), an indication of reduced permeability (*Smith et al., 2020*). Immunoblotting for SFK Y418 phosphorylation, representing c-Src activation, showed a biphasic induction in the control. Treatment with L-NMMA efficiently suppressed the second pY418 peak at 10 min of VEGFA stimulation (*Figure 2—figure supplement 2E*). The inhibition by L-NMMA of the second Y418 peak observed here mirrors results obtained by Li et al., where a leakage-resistant VEGFR2 Y949F mutant also lacked the second pY418 peak in immunoblots (*Li et al., 2016*). Upon in vivo analyses, immunostaining for c-Src pY418 however failed to reveal differences in pY418 c-Src levels between $Nos3^{+/+}$ and $Nos3^{S1176A/S1176A}$ retinas (*Figure 2—figure supplement 3A,B*). These results indicate that the pY418 antibody is not specific to c-Src phosphorylation and likely recognizes several related Src family kinases (SFKs) such as Yes and Fyn. Thus, the double pY418 peaks appearing in the immunoblots may be due to phosphorylation of distinct SFKs.

To overcome the potential lack in antibody specificity for c-Src, we next employed the PLA (*Söderberg et al., 2006*). For this purpose, we used oligonucleotide-ligated secondary antibodies detecting primary antibodies against murine c-Src protein and the conserved (across c-Src, Yes, and Fyn) pY418 residue on endothelial cells isolated from lungs of $Nos^{+/+}$ and $Nos3^{S1176A/S1176A}$ mice and treated or not with VEGFA. The PLA was combined with counterstaining for VE-cadherin (*Figure 2C*). As shown in *Figure 2D*, VEGFA stimulation increased PLA spots, representing pY418 specifically on c-Src, at least fivefold in the isolated endothelial cells (iECs) from $Nos^{+/+}$ mice, but not in iECs from $Nos3^{S1176A/S1176A}$ mice (see *Figure 2—figure supplement 3C* for PLA negative controls). The c-Src pY418 PLA spots co-localized with VE-cadherin immunostaining (*Figure 2E*).

These data indicate that eNOS S1176 phosphorylation and the consequent formation of NO are essential for the accumulation of active c-Src at endothelial junctions to induce the phosphorylation of VE-cadherin at Y685.

## Suppressed vascular leakage in $Nos3^{S1176A/S1176A}$ retinas after OIR-challenge

In the retina, the blood–retina barrier (BRB) controls vascular permeability; however, the BRB is disrupted in retinopathies, causing edema and vision loss (*Klaassen et al., 2013*; *Zhao et al., 2015*). Edema correlates with increased vessel permeability, which is dependent on the phosphorylation status of VE-cadherin (*Wessel et al., 2014*) and c-Src activity (*Wallez et al., 2007*). To assess the role for eNOS specifically in vessel leakage after hypoxia-driven VEGFA production, 25 nm fluorescent microspheres were injected in the tail vein of P17 $Nos3^{+/+}$ and $Nos3^{S1176A/S1176A}$ mice, after OIR-challenge. Confocal image analysis showed accumulation of microspheres outside the vascular tufts, in agreement with enhanced vessel leakage upon OIR (*Figure 3A*). The accumulation of microspheres normalized to tuft area was significantly lower in $Nos3^{S1176A/S1176A}$ retinas compared to $Nos3^{+/+}$ (*Figure 3B*).

## Reduced tuft area as a result of pharmacological inhibition of NO formation

The $Nos3^{S1176A/S1176A}$ mouse is unable to produce NO in the endothelium due to the non-phosphorylatable alanine replacing S1176. As we could not unequivocally exclude that the vascular effects observed in the $Nos3^{S1176A/S1176A}$ mutant were dependent on non-NO synthesis events linked to S1176 phosphorylation, we used L-NMMA to inhibit the formation of NO.

Intraperitoneal injections of L-NMMA (20 µg/g body weight) were given daily to wild-type C57Bl/6J pups during P12–P17 (injections on days P12–P16), thus treatment was initiated before pathological neovessels were established (prevention therapy). This preventive administration had no effect on NG2-positive pericyte coverage or infiltration of CD68-positive macrophages (*Figure 4—figure supplement 1A–D*). However, L-NMMA treatment significantly reduced the area of vascular tufts formed by P17 (*Figure 4A,B*) but did not affect the avascular area (*Figure 4C*). The average tuft size was decreased (*Figure 4D*), while the total number of individual tufts increased with L-NMMA treatment (*Figure 4E*). As smaller tufts can fuse to form larger structures (*Prahst et al., 2020*), the increase in individual tufts in the L-NMMA-treated litter mates may reflect the suppressed growth and fusion of the tufts. The constitutive inhibition of eNOS activation in the $Nos3^{S1176A/S1176A}$ retina resulted in tufts that were more efficiently perfused than in the wild-type retina (*Figure 1—figure*

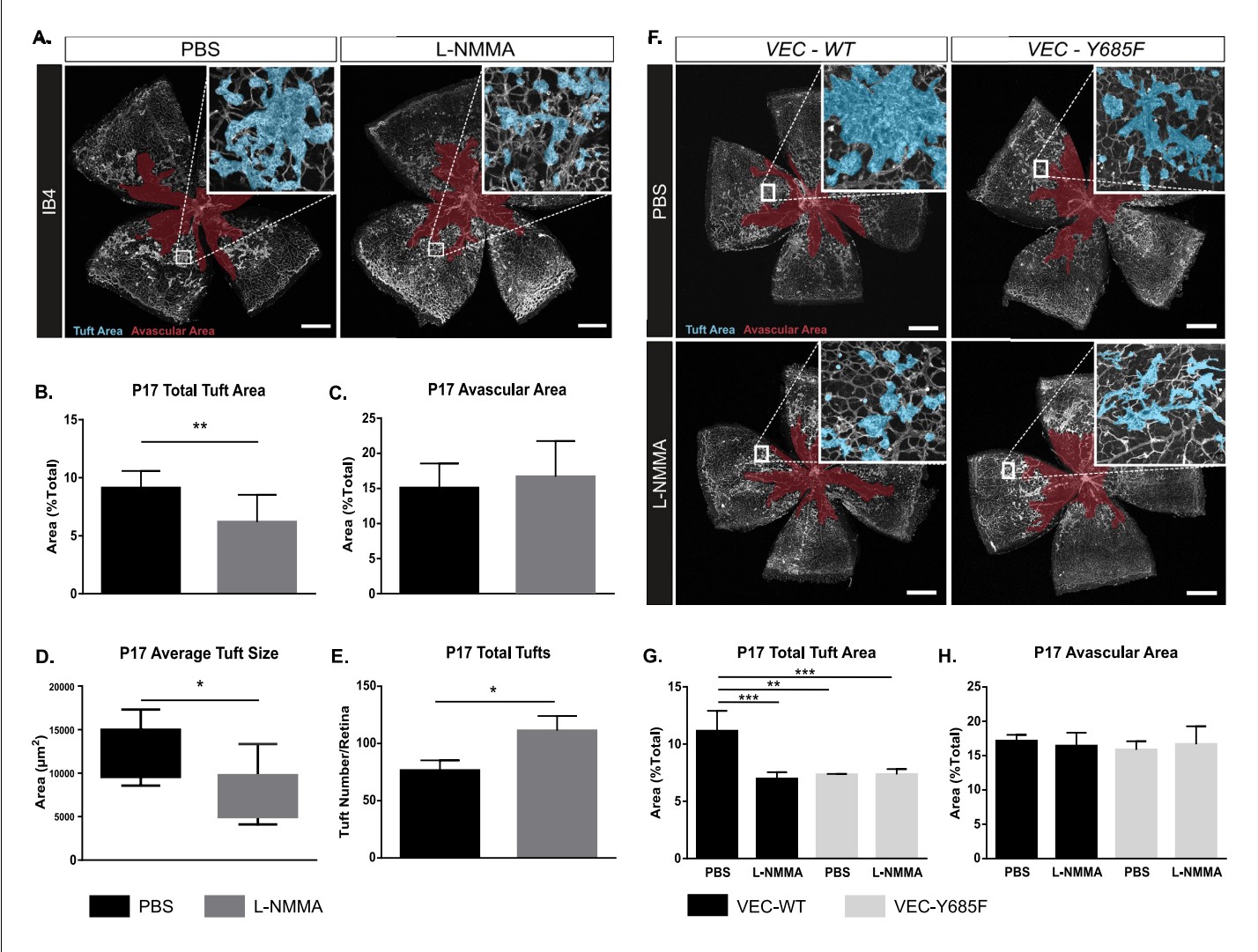

**Figure 4.** OIR-challenged mice treated with NO inhibitor L-NMMA. (**A**) Representative images of whole mount retinas from PBS- and L-NMMA-treated (P12–P16) wild-type C57Bl/6 mice, collected on P17 after OIR-challenge, and stained with isolectin B4 (IB4). Avascular area as a result of OIR is marked in magenta and tufts in blue. Scale bar = 500 μm. (**B, C**) Tuft area and avascular area expressed as percentage of total vascular area at P17. (**D, E**) Tuft size in μm² and total number of tufts/field of vision at P17. Mean ± S.E.M. n = 8 (PBS) and 9 (L-NMMA) treated mice. *, **p<0.05, 0.01; t-test. (**F**) Representative images of whole mount IB4-stained P17 retinas from OIR-challenged VEC-WT and VEC-Y685F mice injected with PBS or L-NMMA during P12-P16. Avascular area is marked in magenta and tufts in blue. Scale bar = 500 μm. (**G**) Tuft area normalized to total vascular area in PBS or L-NMMA-treated VEC-WT and VEC-Y685F retinas. (**H**) Avascular area normalized to total vascular area. Mean ± S.E.M. n = 8 (VEC-WT) and 8 (VEC-Y685F) mice. **, ***p<0.01, 0.001; two-way ANOVA, Sidak's multiple comparison test.

The online version of this article includes the following source data and figure supplement(s) for figure 4:

**Source data 1.** Excel file containing numerical values collected from OIR-induced tuft formation experiments in PBS- and L-NMMA-treated WT retinas shown in *Figure 4* and *Figure 4—figure supplement 1*.

**Figure supplement 1.** Pericytes, macrophage influx, and vascular perfusion in OIR-challenged WT retinas treated with PBS or L-NMMA.

*supplement 3E,F*). In contrast, with L-NMMA treatment, tuft perfusion was equivalent to that in the vehicle-treated pups (*Figure 4—figure supplement 1E,F*). It is expected that constitutive genetic inhibition and pharmacological inhibition during an interval after the hyperoxia treatment would differently suppress tuft formation and growth and result in distinct tuft morphologies, with more or less open lumens.

## NO and VE-cadherin Y685 phosphorylation operate on the same pathway regulating vascular leakage

To further explore the relationship between NO and VE-cadherin pY685 in the formation of leaky, pathological vessels, we used mice expressing wild-type VE-cadherin (VEC-WT) or mutant VE-cadherin wherein phosphorylation at position 685 is abolished by exchanging the tyrosine (Y) for phenylalanine (F), VEC-Y685F (*Wessel et al., 2014*). VEC-Y685F mice show suppressed induction of vascular leakage in the healthy skin (*Wessel et al., 2014*). We hypothesized that if NO modulates vascular leakage and tuft formation via a non-VE-cadherin Y685 pathway, L-NMMA would impart an additional reduction in tuft area to OIR-challenged VEC-Y685F mice. To test whether the VEC-Y685F mouse would respond to NOS inhibition, L-NMMA (20 μg/g body weight) was administered by intraperitoneal injection of VEC-WT and VEC-685F mice during the relative hypoxic period (injections on days P12–P16). Results show that L-NMMA treatment did not further suppress tuft formation in Y685F mice at P17. The reduction in tuft area was similar, about 50%, in PBS and L-NMMA-treated Y685F retinas and comparable to that seen in L-NMMA-treated VEC-WT mice (*Figure 4F,G*). The avascular area remained unaffected by all treatments (*Figure 4H*).

## Single-dose L-NMMA decreases vascular leakage in the retina

We next aimed to more closely mimic a clinical situation by administering L-NMMA to OIR-challenged wild-type mice with established pathological vessels (intervention therapy). Mice were given one injection of L-NMMA (60 μg/g body weight) at P16. At P17, microspheres were injected and after 15 min, the experiment was terminated. The area of extravascular microspheres, assessed after normalization to tuft area (*Figure 5A,B*) or to total microsphere area (*Figure 5A,C*), was reduced by 50–60% after the single-dose treatment with L-NMMA compared to PBS. Of note, the total microsphere area was not affected by the L-NMMA treatment (*Figure 5D*), indicating that microspheres to a large extent were present in the vascular lumen in the L-NMMA-treated condition, while in the PBS control, they had crossed the disrupted barrier to the extravascular space. The tuft area was not affected by the L-NMMA treatment (*Figure 5E*). Analyses of a parallel cohort of mice showed that there was no effect of the single-dose L-NMMA treatment on vessel pericyte coverage, macrophage infiltration, or tuft perfusion (*Figure 5—figure supplement 1A–F*). Thus, these data indicate that leakage could be suppressed even from established neovascular structures.

## Discussion

The results presented here show that eNOS/NO plays a role in the postcapillary venous system by directly affecting the endothelial barrier to exacerbate vascular hyperpermeability in retinopathy (*Figure 6*). According to the consensus model, eNOS-generated NO regulates vascular permeability by inducing the relaxation of perivascular smooth muscle cells. NO produced in endothelial cells diffuses across the vascular wall and activates soluble guanylate cyclase leading to protein kinase G activation in vSMC, lowering cellular $Ca^{2+}$, and promoting vascular relaxation, increased blood flow, and reduced blood pressure (*Surks et al., 1999*). Indeed, vessel dilation is a part of the tissue deterioration seen in diabetic retinopathy (*Bek, 2013*; *Grimm and Willmann, 2012*). However, there is also evidence for a direct role for eNOS and NO in endothelial cells, as constitutive eNOS deficiency inhibits inflammatory hyperpermeability in the mouse cremaster muscle treated with platelet-activating factor (*Hatakeyama et al., 2006*). Moreover, NO can regulate phosphorylation of VE-cadherin in adheren junctions in vitro in microvascular endothelial cell cultures (*Di Lorenzo et al., 2013*). It is likely that these multifaceted effects of eNOS/NO are differently established in different vessel types. eNOS/NO-dependent vessel dilation is dependent on the vSMC coverage in arterioles and arteries; in contrast, adherens junction stability affects mainly the postcapillary venular bed (*Orsenigo et al., 2012*). In the skin, prevenular capillaries and postcapillary venules, with sparse vSMC coverage, respond to VEGFA with increased paracellular permeability, while arterioles/arteries do not (*Honkura et al., 2018*). These distinctions are important as exaggerated and chronic VEGFA-driven paracellular permeability in disease leads to edema and eventually to tissue destruction (*Nagy et al., 2012*). Therefore, we explored the consequences of NO deficiency using a mutant eNOS mouse model, with a serine to alanine exchange at 1176, as well as treatment with the

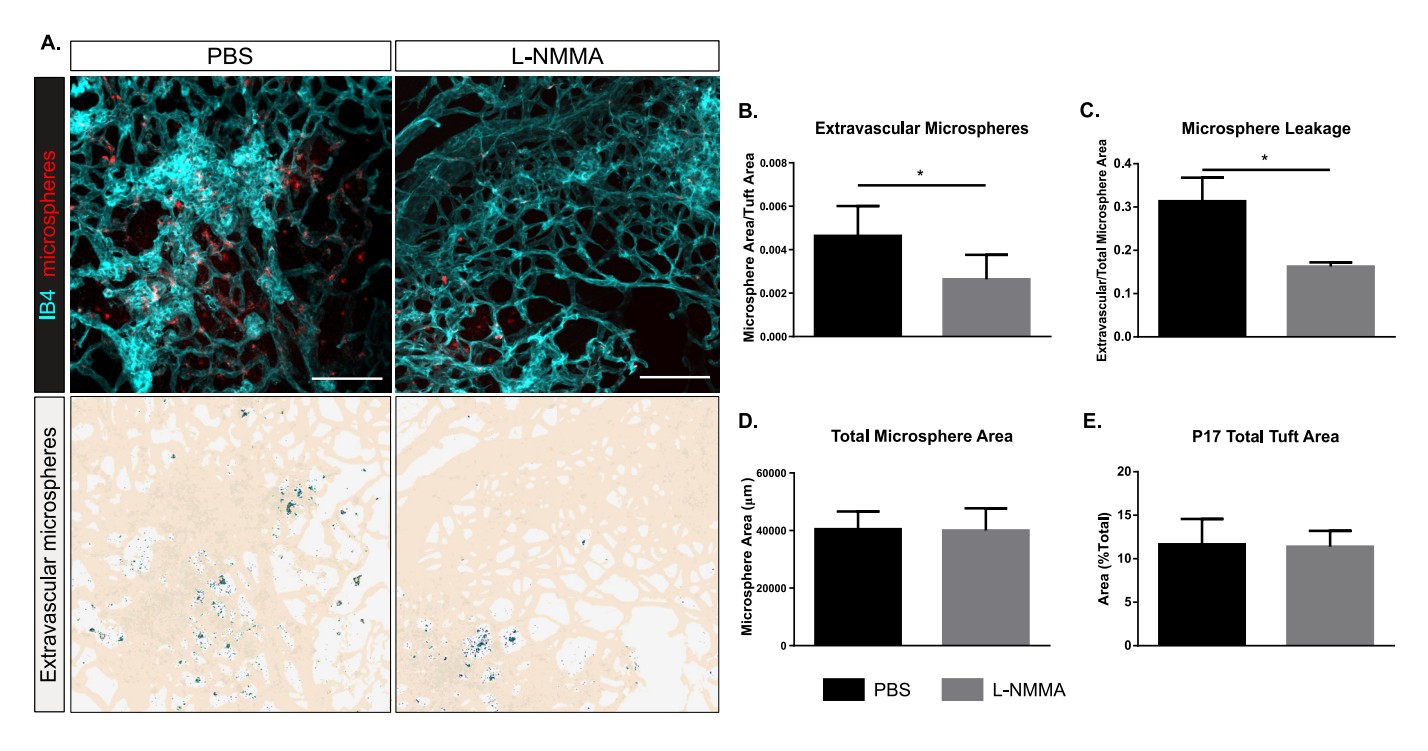

**Figure 5.** Decreased leakage from retinal vascular tufts after single-dose L-NMMA treatment. (**A**) Representative images of tufts from *Nos3$^{+/+}$* (wild type) mice treated with PBS or L-NMMA (60 µg/g body weight) 24 hr before tail-vein injection of 25 nm microspheres. Retinas were stained using isolectin B4 (IB4; cyan), microspheres (red) appear in and around the tufts. Scale bar = 100 µM. Lower panels show leakage maps. Dots that do not overlap with vessels (beige) are considered extravascular. (**B**) Quantification of the average area of extravascular microspheres normalized to IB4 area. (**C**) Quantification of the average area of extravascular microspheres normalized to total microsphere area. (**D**) Quantification of average total microsphere area in PBS- or L-NMMA-treated wild-type mouse retinas. (**E**) Quantification of tuft area normalized to total vascular area. Mean ± S.E.M. n = 5 (PBS) and 5 (L-NMMA) treated mice. *p<0.05; t-test.

The online version of this article includes the following source data and figure supplement(s) for figure 5:

**Source data 1.** Excel file containing numerical values collected from microsphere leakage analyses from OIR retinas in single-treated PBS and L-NMMA WT mice shown in *Figure 5*, *Figure 5—figure supplement 1*.

**Figure supplement 1.** Pericytes, macrophage influx, and vascular perfusion in OIR-challenged WT retinas treated with a single injection of PBS or L-NMMA.

NO inhibitor L-NMMA, to examine how attenuating NO production affects VEGFA-dependent pathological angiogenesis in ocular disease.

Our data shows that while the attenuation of eNOS S1176 phosphorylation was dispensable for vascular development in the retina and for endothelial survival during hyperoxia, growth of pathological vessel tufts in the subsequent phase of relative hypoxia was suppressed, while pericyte coverage and inflammation remained unaffected. Also, treatment with L-NMMA during the hypoxic phase reduced growth of vascular tufts in the retina. In agreement with earlier literature, we conclude that eNOS activity and NO formation influence pathological angiogenesis in the eye (*Ando et al., 2002*; *Brooks et al., 2001*; *Edgar et al., 2012*). Of note, tufts that were established in the *Nos3$^{S1176A/S1176A}$* mice leaked less, in spite of similar levels of VEGFA being produced as in the wild-type retina. In an attempt to mimic the clinical situation, we treated P16 mice with established vascular tufts with a single dose of L-NMMA. At the examination 24 hr later, tuft area remained unaffected, while the leakage of 25 nm microspheres was reduced by 50–60%. Whether leakage regulation is separable from the regulation of growth of new vessels has been a matter of debate. Pathological angiogenesis in the retina is intimately associated with leakage and edema (*Smith et al., 2020*). Exactly how junction stability plays a role in the neoangiogenic process is unclear. However, leakage and the production of a provisional matrix is postulated to be a prerequisite for the growth of angiogenic sprouts (reviewed in *Nagy et al., 2012*). With the effects of the L-NMMA intervention treatment

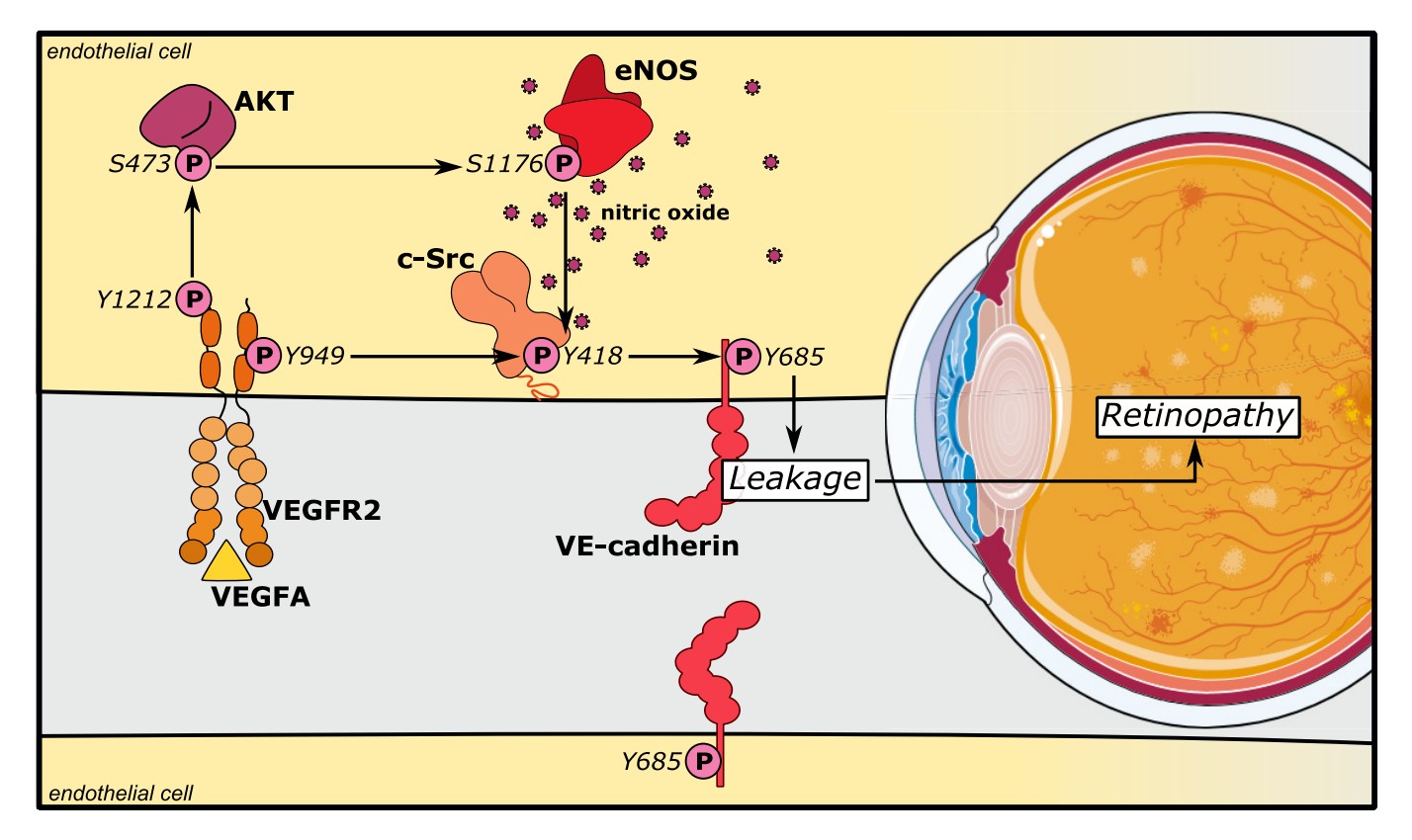

**Figure 6.** eNOS/NO modulates VE-cadherin Y685 phosphorylation via c-Src in a VEGFA/VEGFR2-dependent manner. VEGFA through VEGFR2 and its phosphosite Y1212 induces a chain of consecutive reactions in endothelial cells: phosphorylation of AKT at S473 and eNOS at S1176. The VEGFR2 phosphosite Y949 mediates phosphorylation of c-Src at Y418 and of VE-cadherin at Y685. Combined, these activating phosphorylation reactions disrupt the vascular barrier by dissociating VE-cadherin's homophilic interactions, resulting in macromolecular leakage. eNOS/NO regulates activation of c-Src to enhance VE-cadherin Y685 phosphorylation and internalization.

shown here, we can conclude that these responses indeed can be separated, at least in the short term.

L-NMMA and its analog, L-NAME, induce vasoconstriction by halting the NO/soluble guanylyl cyclase signaling pathway and consequent vasodilation (reviewed in *Ahmad et al., 2018*; *Thoonen et al., 2013*). Thus, a daily intake of L-NAME in rats (40–75 µg/g/day for 4–8 weeks) leads to vasoconstriction (*Ribeiro et al., 1992*; *Simko et al., 2018*; *Vrankova et al., 2019*; *Zanfolin et al., 2006*). NOS inhibitors have also been used clinically, for example, by administration of L-NMMA to increase the mean arterial pressure in cardiogenic shock. In a typical regimen, 1 mg.kg$^{-1}$.hr$^{-1}$ of L-NMMA is administered over 5 hr (*Cotter et al., 2000*). While we cannot exclude an effect on vaso-constriction with the single IP injection of L-NMMA used here (60 µg/g), blood flow appeared unaf-fected as equal amounts of microspheres arrived in the retinal vasculature, while leakage into the extravascular space was substantially reduced in the L-NMMA treated mice compared to the con-trols. The leakage suppression did not involve changes in pericyte coverage of vessels or in the degree of inflammation. While perfusion was enhanced in tufts that still formed in the $Nos3^{S1176A/S1176A}$ retinas after OIR-challenge, there was no effect on perfusion in tufts remaining after multiple- or single-dose L-NMMA treatment, possibly reflecting differences in tuft morphology and the degree of vessel patency.

Mechanistically, our results place c-Src downstream of eNOS activity. c-Src has also been placed upstream of eNOS by c-Src's regulation of the PI3K-AKT pathway, of importance for eNOS activa-tion (*Haynes et al., 2003*). VEGFR2-induced activation of c-Src has however been mapped to the pY949 residue in VEGFR2 (*Li et al., 2016*), while activation of Akt is dependent on Y1212

(*Testini et al., 2019*). In VEGFA-stimulated endothelial cells, c-Src is implicated in phosphorylation of VE-cadherin at Y685 and potentially other tyrosine residues, to induce vascular permeability (*Orsenigo et al., 2012*; *Wallez et al., 2007*). The fact that the mutant VEC-Y685F mice in which the Y685 residue has been replaced with a non-phosphorylatable phenylalanine were resistant to L-NMMA inhibition, indicates that the VEGFR2 Y1212/eNOS/NO pathway acts on adherens junction through c-Src (*Li et al., 2016*). Combined, our data support a model in which eNOS phosphorylation at S1176 is required for the activation of c-Src, which in turn phosphorylates VE-cadherin at Y685, inducing transient disintegration of adherens junctions and increased paracellular permeability (*Figure 6*). These in vivo results potentially provide meaningful mechanistic and therapeutic insights with regard to retinopathy of prematurity (*Cunha-Vaz et al., 2011*).

A considerable challenge in the analysis of c-Src activity is the close structural relatedness between the kinase domains of c-Src and the related SFKs Yes and Fyn (*Sato et al., 2009*), which are also expressed in endothelial cells. The amino acid sequence covering the activating tyrosine is entirely conserved between these three cytoplasmic tyrosine kinases such that the antibodies against pY418 c-Src in fact reacts with all three SFKs. In agreement, immunoblotting using antibodies against pY418 c-Src resulted in biphasic peaks of activity possibly reflecting activation of different SFKs. Moreover, pY418 immunostaining of wild-type and *Nos3^{S1176A/S1176A}* retinas after OIR failed to reveal a dependence on eNOS catalytic activity for activation of c-Src, in agreement with the findings of *Di Lorenzo et al., 2013*. However, combining oligonucleotide-linked secondary antibodies reacting with c-Src protein and pY418 Src in a PLA on isolated endothelial cells demonstrated a critical role for eNOS activity on accumulation of active, pY418-positive c-Src at endothelial junctions. The question remains how eNOS/NO influences c-Src activity? NO can couple with cysteine thiols to form S-nitroso-thiols, which may affect the folding and function of the target protein. In accordance, Rahman et al. demonstrated S-nitrosylation of c-Src at the kinase domain cysteine 498, correlating with increased c-Src activity (*Rahman et al., 2010*).

Although not experimentally addressed in this study, retinal diseases, such as diabetic retinopathy and adult macular degeneration, are accompanied by excessive permeability (*Antonetti et al., 2012*; *Campochiaro, 2015*). Of note, patients with diabetic retinopathy have elevated levels of NO in the aqueous humor, and particular eNOS polymorphisms are associated with protection or increased risk for diabetic retinopathy (for a review, see *Opatrilova et al., 2018*) and macular edema (*Awata et al., 2004*). Our results show that pharmacological inhibition of NO production after OIR challenge in mice can prevent vascular leakage. Thus, NO inhibitors applied in combination with anti-VEGF therapy, delivered in low but still efficient doses, could possibly be of clinical benefit. Local delivery of NOS inhibitors may be needed to avoid drawbacks with systemic delivery, such as hypertension, or any other vasoconstriction-associated adverse events. A limitation of the findings reported here was that neither the multiple-dose nor single-dose L-NMMA treatment improved perfusion of remaining tufts after OIR challenge. However, as tuft formation and leakage were suppressed with the multiple-dose treatment, a relative larger fraction of vessels remained healthy, with normal perfusion.

In conclusion, the data presented here establish a critical role for eNOS in endothelial cells, regulating c-Src activation downstream of VEGFA/VEGFR2, and thereby, in VE-cadherin-regulated endothelial junction stability and vascular leakage in retinal pathology.

## Acknowledgements

We gratefully acknowledge the expert assistance of Pernilla Martinsson, Uppsala University.

## Additional information

### Funding

| Funder | Grant reference number | Author |
| --- | --- | --- |
| Swedish Cancer Foundation | 19 0119 Pj 01 H | Lena Claesson-Welsh |
| Vetenskapsrådet | 2020-01349 | Lena Claesson-Welsh |
| Knut och Alice Wallenbergs | KAW 2020.0057 | Lena Claesson-Welsh |

| | | |
|---|---|---|
| Stiftelse | | |
| Knut och Alice Wallenbergs Stiftelse | KAW 2015.0275 | Lena Claesson-Welsh |
| Fondation Leducq | 17 CVD 03 | Lena Claesson-Welsh |
| National Institutes of Health | R35 HL 139945 | William C Sessa |
| National Institutes of Health | P01HL 1070205 | William C Sessa |
| American Heart Association | Merit Award | William C Sessa |
| Deutsche Forschungsge-meinschaft | SFB1450 B03 | Dietmar Vestweber |
| Deutsche Forschungsge-meinschaft | CRU342 P2 | Dietmar Vestweber |

The funders had no role in study design, data collection and interpretation, or the decision to submit the work for publication.

### Author contributions

Takeshi Ninchoji, Conceptualization, Formal analysis, Validation, Investigation, Visualization, Writing - review and editing; Dominic T Love, Conceptualization, Formal analysis, Validation, Investigation, Visualization, Methodology, Writing - review and editing; Ross O Smith, Supervision, Methodology, Writing - review and editing; Marie Hedlund, Investigation, Writing - review and editing; Dietmar Vestweber, William C Sessa, Resources, Writing - review and editing; Lena Claesson-Welsh, Conceptualization, Resources, Supervision, Funding acquisition, Writing - original draft, Project administration, Writing - review and editing

### Author ORCIDs

Dominic T Love https://orcid.org/0000-0001-8530-1352
Ross O Smith http://orcid.org/0000-0003-4239-3204
Dietmar Vestweber http://orcid.org/0000-0002-3517-732X
William C Sessa http://orcid.org/0000-0001-5759-1938
Lena Claesson-Welsh https://orcid.org/0000-0003-4275-2000

### Ethics

Animal experimentation: Mouse husbandry and oxygen-induced retinopathy (OIR) challenge took place at Uppsala University, and the University board of animal experimentation approved all animal work for the studies (ethical permit 5.2.18-8927/16). Animal handling was in accordance to the ARVO statement for the Use of Animals in Ophthalmologic and Vision Research. Professional animal care was provided and overseen by University veterinarians. Every effort was made to minimize suffering of the animal.

### Decision letter and Author response

Decision letter https://doi.org/10.7554/eLife.64944.sa1
Author response https://doi.org/10.7554/eLife.64944.sa2

## Additional files

### Supplementary files

• Supplementary file 1. Body weights for OIR experiments.

• Transparent reporting form

### Data availability

All data generated or analysed during this study are included in the manuscript and supporting files. Source data files have been provided for Figures 1–5. Source data files have also been deposited with Dryad: https://doi.org/10.5061/dryad.x69p8czhv.

The following dataset was generated:

| Author(s) | Year | Dataset title | Dataset URL | Database and Identifier |
|---|---|---|---|---|
| Love D, Ninchoji T, Claesson-Welsh L | 2021 | eNOS/NO and their role in modifying the vascular barrier in retinopathy - Source data | https://doi.org/10.5061/dryad.x69p8czhv | Dryad Digital Repository, 10.5061/dryad.x69p8czhv |

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
