## [Decision Letter]

**Acceptance summary:**

The strength of this study is to clarify the roles of eNOS in ischemic neovascularization and vessel permeability using an oxygen-induced retinopathy (OIR) mouse model. It shows that either systemic NO formation inhibitor L-NMMA or eNOS (S1176A) mutant causes reduced retinal vascular tuft formation, and also reduces vascular leakage by stabilizing VE-cadherin in a c-Src-dependent manner. It also shows that a single dose of L-NMMA restores the vascular barrier and prevents leakage. Based on these findings, they have concluded that eNOS induces destabilization of adherens junctions and vascular hyperpermeability by converging with the VEGFA/VEGFR2/c43 Src/VE-cadherin pathway and that this pathway can be selectively inhibited by blocking NO formation. This study is well-designed and well-presented, and mechanistically solid.

**Decision letter after peer review:**

Thank you for submitting your article "eNOS-induced vascular barrier disruption in retinopathy by c-Src activation and tyrosine phosphorylation of VE-cadherin" for consideration by *eLife*. Your article has been reviewed by 2 peer reviewers, including Gou Young Koh as the Reviewing Editor and Reviewer #1, and the evaluation has been overseen by Didier Stainier as the Senior Editor.

The reviewers have discussed the reviews with one another and the Reviewing Editor has drafted this decision to help you prepare a revised submission.

We would like to draw your attention to changes in our policy on revisions we have made in response to COVID-19 (https://elifesciences.org/articles/57162). Specifically, when editors judge that a submitted work as a whole belongs in *eLife* but that some conclusions require a modest amount of additional new data, as they do with your paper, we are asking that the manuscript be revised to either limit claims to those supported by data in hand, or to explicitly state that the relevant conclusions require additional supporting data.

The comments of all two reviewers are in good agreement. While the reviewers found this study is well-designed and well-presented, mechanistically solid, and to be of importance as a translational research, they raised concerns about the limitation to be generalized to apply major ocular diseases including wet-AMD and diabetic retinopathy and the inconsistencies of the results and interpretations to the previous reports. The authors are required to carefully address the comments point-by-point in a data-driven manner or with further analyses or discussions. Specifically, the authors are encouraged to pay attention to the major comments 2 and 3 of reviewer 1 and comments 1 and 2 of reviewer 2. If necessary, please provide the reasons for not implementing the suggested changes.

Reviewer #1:

This study has elegantly clarified the role of eNOS in ischemic neovascularization and vessel permeability using an oxygen-induced retinopathy (OIR) mouse model. The authors show that either systemic NO formation inhibitor L-NMMA or eNOS (S1176A) mutant causes reduced retinal vascular tuft formation, and also reduces vascular leakage by stabilizing VE-cadherin in a c-Src-dependent manner. Because the OIR model can reflect only "retinopathy of prematurity" in premature newborns but cannot fully reflect wet-AMD (in fact, choroidal vessel disease) and diabetic retinopathy in adult, this study has a limitation to be generalized to apply major ocular diseases including wet-AMD and diabetic retinopathy. Nevertheless, this study is well-designed and well-presented, and mechanistically solid, and thus it deserves to be published in *eLife*.

The study conducted by Ninchoji et al. has clarified the role of eNOS in ischemic neovascularization and vessel permeability using an oxygen-induced retinopathy (OIR) mouse model. They show that either systemic NO formation inhibitor L-NMMA or eNOS (S1176A) mutant causes reduced retinal vascular tuft formation, and also reduces vascular leakage by stabilizing VE-cadherin in a c-Src-dependent manner. They also show that a single dose of L-NMMA restores the vascular barrier and prevents leakage. Based on these findings, they have concluded that eNOS induces destabilization of adherens junctions and vascular hyperpermeability by converging with the VEGFA/VEGFR2/c43 Src/VE-cadherin pathway and that this pathway can be selectively inhibited by blocking NO formation. This study is well-designed and well-presented, and mechanistically solid. However, the following comments need to be appropriately addressed:

1. The OIR model can reflect "retinopathy of prematurity" in premature newborns but cannot fully reflect wet-AMD (in fact, choroidal vessel disease) and diabetic retinopathy in adult. Therefore, the authors need to tone down the generalization and limit the applicability.

2. Pericyte dropout and infiltration of macrophages are critical components to vascular leakage in the OIR model, as shown in recent papers (Park et al., Nature Communication, 2017; Ogura et al., JCI Insight 2017; Omori et al., JCI Insight 2018). What are the changes in PC-EC coupling and macrophage infiltration in either L-NMMA treatment or eNOS(S1176A) mutant by OIR.

3. Because the vascular tuft and deformed vessels formed by OIR have generally no circulation, it causes severe hypoxia or anoxia to the retinal neurons. Is there restoration in the retinal circulation after L-NMMA treatment? It can be measured by a lectin perfusion assay.

4. Restoration of functional blood vessels in the avascular area is the most important point for treating the patients of "retinopathy of prematurity". In this regard, single treatment of NMMA is not sufficient. Therefore, the description of this therapeutic option needs to be further defined.

Reviewer #3:

Ninchoji T. et al. demonstrated the signaling cascade that worsens the endothelial cell (EC) barrier function by phosphorylating vascular endothelial cadherin (VE-cadherin). The effect of endothelial NO synthetase (eNOS, Nos3) on barrier function has been different in different vessel types and in different disease models. In addition, it is also controversial whether c-Src is upstream or downstream of eNOS. The authors delineate the signaling cascade that regulates retinal barrier function using the mice (genetically mutated Nos3 (S1176A) that cannot be phosphorylated) and the mice (VEC-Y685F) in which VE-cadherin cannot be phosphorylated in oxygen-induced retinopathy (OIR) model.

The authors extended their previous results reported in *eLife* (Smith et al. *eLife* 2020;9: e54056) to show the VEGF-dependent vascular leakage in OIR model. Previously, they reported that phosphorylation of VE-cadherin Y685 results in vessel leakage. In the present study, they analyzed Tuft area and avascular area similarly to the previous report and delineated the VEGF/VEGFR2-dependent signaling that causes vascular leakage. In the mice in which Nos3S1176A was knocked-in, vascular leakage was suppressed. Moreover, VEC-Y685F mutant mice showed less tuft area than wildtype. This is consistent with their previous result. The reduction in tuft area was comparable in PBS and L-NMMA-treated Y685F mouse retinas and that seen in L-NMMA-treated wildtype mice. The involvement of C-Src downstream of VEGF/VEGFR2-eNOS was shown in proximity ligation assay (PLA). Collectively, they delineate the signaling mediated by VEGF/VEGFR2-eNOS (S1176 phosphorylation)-c-Src (Y418 phosphorylation)-VE-cad(Y685 phosphorylation) that induces leakage in OIR model.

The signaling cascade is proven by the two different knock-iin mice (Nos3 S1176A and VEC Y685F) in the established OIR model in vivo, while other groups show eNOS-dependent barrier stabilization effect on ECs in different model.

1. Involvement of c-Src in OIR

There is one inconsistency in the model illustrated in Figure 6. In the previous report by Smith (the same group), the authors mentioned that pY949 is VEGFR2 appeared not to contribute to regulation of c-Src kinase activity in the retina vasculature. However, the model in Figure 6 clearly points to the Y949 of VEGFR2 and c-Src Y418. Considering the previous results and the new results, the authors need to revise the involvement of c-Src in OIR model Immunoblot analyses of in vitro assay using human retinal microvascular endothelial cells might help the readers to understand the upstream and downstream of VEGFR2; namely AKT-eNOS-c-Src and another direct c-Src-mediated VE-cad phosphorylation by showing time course of phosphorylation of Akt and eNOS. In addition to this, c-Src phosphorylation might be analyzed in a time-dependent manner, although Fyn or Yes might be similarly phosphorylated. At least, time course can be analyzed.

2. Anti-VEGF therapy

The authors describe the disadvantage of anti-VEGF therapy in the introduction. However, the vascular leakage in OIR is dependent on VEGF/VEGFR2 as illustrated in Figure 6. Why is anti-VEGF therapy ineffective in OIR? VEGF-dependent eNOS activation should be inhibited by ant-VEGF therapy. Because L-NMMA might become a potential candidate as an ant-eNOS therapy, the authors had better mention the usefulness of this as an alternative or additional therapy for retinopathy.

3. The intensity of red in Figure 2 -supplement figure 2c is too low. These panels should be replaced.

4. L-NMMA should be explained when it appears first in page 9 instead of page 11

---

## [Author Response]

Reviewer #1:This study has elegantly clarified the role of eNOS in ischemic neovascularization and vessel permeability using an oxygen-induced retinopathy (OIR) mouse model. The authors show that either systemic NO formation inhibitor L-NMMA or eNOS (S1176A) mutant causes reduced retinal vascular tuft formation, and also reduces vascular leakage by stabilizing VE-cadherin in a c-Src-dependent manner. Because the OIR model can reflect only "retinopathy of prematurity" in premature newborns but cannot fully reflect wet-AMD (in fact, choroidal vessel disease) and diabetic retinopathy in adult, this study has a limitation to be generalized to apply major ocular diseases including wet-AMD and diabetic retinopathy. Nevertheless, this study is well-designed and well-presented, and mechanistically solid, and thus it deserves to be published in eLife.The study conducted by Ninchoji et al. has clarified the role of eNOS in ischemic neovascularization and vessel permeability using an oxygen-induced retinopathy (OIR) mouse model. They show that either systemic NO formation inhibitor L-NMMA or eNOS (S1176A) mutant causes reduced retinal vascular tuft formation, and also reduces vascular leakage by stabilizing VE-cadherin in a c-Src-dependent manner. They also show that a single dose of L-NMMA restores the vascular barrier and prevents leakage. Based on these findings, they have concluded that eNOS induces destabilization of adherens junctions and vascular hyperpermeability by converging with the VEGFA/VEGFR2/c43 Src/VE-cadherin pathway and that this pathway can be selectively inhibited by blocking NO formation. This study is well-designed and well-presented, and mechanistically solid. However, the following comments need to be appropriately addressed:1. The OIR model can reflect "retinopathy of prematurity" in premature newborns but cannot fully reflect wet-AMD (in fact, choroidal vessel disease) and diabetic retinopathy in adult. Therefore, the authors need to tone down the generalization and limit the applicability.

We thank the reviewer for the appreciative comments. We concur with the reviewer’s description of the limitation of the OIR model and have toned down the generalization and applicability. See track changes in the discussion, lines 664-666, 675-679.

2. Pericyte dropout and infiltration of macrophages are critical components to vascular leakage in the OIR model, as shown in recent papers (Park et al., Nature Communication, 2017; Ogura et al., JCI Insight 2017; Omori et al., JCI Insight 2018). What are the changes in PC-EC coupling and macrophage infiltration in either L-NMMA treatment or eNOS(S1176A) mutant by OIR.

We have now performed new experiments and measured NG2-positive pericyte coating and the presence of CD68-positive macrophages in retinas after OIRchallenge of wildtype or *Nos3^S1176A/Nos3S1176A^* mutant mice, as well as mice treated with multiple or single dose of L-NMMA. The extent of pericyte coating and macrophage infiltration was the same in all conditions. These data are now shown in Figure 1—figure supplement 3, Figure 4—figure supplement 1 and Figure 5—figure supplement 1.

3. Because the vascular tuft and deformed vessels formed by OIR have generally no circulation, it causes severe hypoxia or anoxia to the retinal neurons. Is there restoration in the retinal circulation after L-NMMA treatment? It can be measured by a lectin perfusion assay.

We have now performed perfusion using fluorescent lectin and although perfusion is increased in tufts formed in the *Nos3^S1176A/Nos3S1176A^* mutant mice, there was no change in perfusion of tufts formed after multiple or single-dose treatment with L-NMMA (see Figure 4—figure supplement 1 and Figure 5—figure supplement 1). We have added a comment on the lack of improved perfusion in the discussion, lines 625-628.

4. Restoration of functional blood vessels in the avascular area is the most important point for treating the patients of "retinopathy of prematurity". In this regard, single treatment of NMMA is not sufficient. Therefore, the description of this therapeutic option needs to be further defined.

We thank the reviewer for this comment and we of course agree. The single-dose treatment was a useful strategy in distinguishing the effects of the LNMMA treatment on tuft formation (which was unchanged with the single dose) and leakage (which was suppressed). The clinical implication of the L-NMMA treatment we agree was overemphasized and has now been toned down throughout.

Reviewer #3:Ninchoji T. et al. demonstrated the signaling cascade that worsens the endothelial cell (EC) barrier function by phosphorylating vascular endothelial cadherin (VE-cadherin). The effect of endothelial NO synthetase (eNOS, Nos3) on barrier function has been different in different vessel types and in different disease models. In addition, it is also controversial whether c-Src is upstream or downstream of eNOS. The authors delineate the signaling cascade that regulates retinal barrier function using the mice (genetically mutated Nos3 (S1176A) that cannot be phosphorylated) and the mice (VEC-Y685F) in which VE-cadherin cannot be phosphorylated in oxygen-induced retinopathy (OIR) model.The authors extended their previous results reported in eLife (Smith et al. eLife 2020;9: e54056) to show the VEGF-dependent vascular leakage in OIR model. Previously, they reported that phosphorylation of VE-cadherin Y685 results in vessel leakage. In the present study, they analyzed Tuft area and avascular area similarly to the previous report and delineated the VEGF/VEGFR2-dependent signaling that causes vascular leakage. In the mice in which Nos3S1176A was knocked-in, vascular leakage was suppressed. Moreover, VEC-Y685F mutant mice showed less tuft area than wildtype. This is consistent with their previous result. The reduction in tuft area was comparable in PBS and L-NMMA-treated Y685F mouse retinas and that seen in L-NMMA-treated wildtype mice. The involvement of C-Src downstream of VEGF/VEGFR2-eNOS was shown in proximity ligation assay (PLA). Collectively, they delineate the signaling mediated by VEGF/VEGFR2-eNOS (S1176 phosphorylation)-c-Src (Y418 phosphorylation)-VE-cad(Y685 phosphorylation) that induces leakage in OIR model.The signaling cascade is proven by the two different knock-iin mice (Nos3 S1176A and VEC Y685F) in the established OIR model in vivo, while other groups show eNOS-dependent barrier stabilization effect on ECs in different model.1. Involvement of c-Src in OIRThere is one inconsistency in the model illustrated in Figure 6. In the previous report by Smith (the same group), the authors mentioned that pY949 is VEGFR2 appeared not to contribute to regulation of c-Src kinase activity in the retina vasculature. However, the model in Figure 6 clearly points to the Y949 of VEGFR2 and c-Src Y418. Considering the previous results and the new results, the authors need to revise the involvement of c-Src in OIR model Immunoblot analyses of in vitro assay using human retinal microvascular endothelial cells might help the readers to understand the upstream and downstream of VEGFR2; namely AKT-eNOS-c-Src and another direct c-Src-mediated VE-cad phosphorylation by showing time course of phosphorylation of Akt and eNOS. In addition to this, c-Src phosphorylation might be analyzed in a time-dependent manner, although Fyn or Yes might be similarly phosphorylated. At least, time course can be analyzed.

We have followed the reviewer’s suggestion and now provide a time course where the effects of L-NMMA treatment on VEGFA-induced phosphorylation of eNOS, AKT, VE-cadherin and Src have been examined (see Figure 2 – suppl figure 1, lines 452-464). Interestingly, VEGFA induces a biphasic pY418 signal and one of them, the second peak, is inhibited by L-NMMA. We suggest that the different peaks may represent activation of different Src family kinases as the pY418 antibody does not distinguish between c-Src, Yes and Fyn. We therefore would like to keep the PLA results, which allows a firm conclusion on the effect of L-NMMA specifically on c-Src activation. Of note, a PLA signal will be generated when the secondary antibodies detect the primary antibodies (against c-Src protein and against pY418) in close proximity, i.e. when c-Src is phosphorylated on Y418. We have gone through the text and figures to ensure we have described this clearly, for example in Figure 4C, which was erroneously labeled before. We apologize for this oversight.

Taken together, we are not yet ready to revise the involvement of c-Src in the OIR model as the PLA results show activation of c-Src in an eNOS dependent manner at endothelial junctions. To bring this further, we need even more refined tools. Indeed, detailed analyses of the specific biological roles of Src, Yes and Fyn in endothelial cell-specific deletion mouse models have been initiated in our group, but these are challenging projects in their own rights.

2. Anti-VEGF therapyThe authors describe the disadvantage of anti-VEGF therapy in the introduction. However, the vascular leakage in OIR is dependent on VEGF/VEGFR2 as illustrated in Figure 6. Why is anti-VEGF therapy ineffective in OIR? VEGF-dependent eNOS activation should be inhibited by ant-VEGF therapy. Because L-NMMA might become a potential candidate as an ant-eNOS therapy, the authors had better mention the usefulness of this as an alternative or additional therapy for retinopathy.

The standard clinical treatment by now is anti-VEGF therapy and we describe the limitation of this therapy in the introduction. The problem is not necessarily that it is ineffective but rather that it has side effects by targeting VEGFR2 on endothelial cells as well as on neurons. See Introduction, lines 77-83. As suggested by Reviewer 1, we have now toned down the clinical perspective which we agree was overemphasized. However, please see the discussion, lines 672-673, where the potential usefulness of combinatorial treatment with anti-VEGF and NO suppressive drugs was already mentioned.

3. The intensity of red in Figure 2 -supplement figure 2c is too low. These panels should be replaced.

Figure 2 -suppl figure 2c show the PLA controls and these were negative – no signal. The images are representative and have been kept.

4. L-NMMA should be explained when it appears first in page 9 instead of page 11

Amended.